# Nacala-Roof-Material: Drone Imagery for Roof Detection, Classification, and Segmentation to Support Mosquito-borne Disease Risk Assessment

## Abstract

As low-quality housing and in particular certain roof characteristics are associated with an increased risk of malaria, classification of roof types based on remote sensing imagery can support the assessment of malaria risk and thereby help prevent the disease. To support research in this area, we release the Nacala-Roof-Material dataset, which contains high-resolution drone images from Mozambique with corresponding labels delineating houses and specifying their roof types. The dataset defines a multi-task computer vision problem, comprising object detection, classification, and segmentation. In addition, we benchmarked various state-of-the-art approaches on the dataset. Canonical U-Nets, YOLOv8, and a custom decoder on pretrained DINOv2 served as baselines. We show that each of the methods has its advantages but none is superior on all tasks, which highlights the potential of our dataset for future research in multi-task learning. While the tasks are closely related, accurate segmentation of objects does not necessarily imply accurate instance separation, and vice versa. We address this general issue by introducing a variant of the deep ordinal watershed (DOW) approach that additionally separates the interior of objects, allowing for improved object delineation and separation. We show that our DOW variant is a generic approach that improves the performance of both U-Net and DINOv2 backbones, leading to a better trade-off between semantic segmentation and instance segmentation.

## 1 Introduction

Mosquito-borne diseases refer to a group of infectious illnesses transmitted by the bite of mosquitoes. Malaria is a mosquito-borne disease caused by single-celled parasites of the Plasmodium group spread through bites of infected female Anopheles mosquitoes. It ranks among the world's most severe public health problems and is a leading cause of mortality and disease in many developing countries. It is therefore crucial to improve prevention, control, and surveillance measures of malaria, particularly in sub-Saharan Africa (Venkatesan, 2024; WHO, 2023). Low-quality housing built of natural materials, for example, having a thatched roof of grass or palm and having cane, grass, shrub, or mud as internal and external walls, is associated with an increased risk of malaria infection (Dlamini et al., 2017). Sub-standard housing has more mosquito entry points and most malaria transmissions in sub-Saharan Africa occur inside dwellings while the inhabitants are asleep (Tusting et al., 2020; 2017; Jatta et al., 2018; Tusting et al., 2019). Houses with metal roofs are hotter in the daytime than houses with thatched roofs. This may reduce mosquito survival and inhibit parasite development within the mosquito in metal roof houses. On this basis, the proliferation of modern construction materials in sub-Saharan Africa may have contributed decisively to the reduction of malaria cases (Tusting et al., 2019). Classification of roof characteristics thus holds potential to support malaria surveillance and control programs. Roof characteristics, such as geometry, material, and condition can be monitored using remote sensing imagery to advance risk assessment of mosquito-borne diseases and guide mitigation strategies, especially when detailed health and socioeconomic data are scarce.

Here, we introduce the Nacala-Roof-Material drone-imagery dataset to support the development of machine learning algorithms for automated building *and* roof type mapping in low-income areas

prone to malaria risk. Our dataset is based on high-resolution drone imagery ($\approx 4.4\,\mathrm{cm}$) of peri-urban and rural settlements in Nacala, Mozambique. An NGO (anonymous during peer review) delineated 17954 buildings and categorized them according to five roof types, and the authors again carefully verified all annotations.

We define three tasks on the Nacala-Roof-Material dataset, building detection, multi-class roof type classification, and pixel-level building segmentation. While these tasks are related, closer inspection reveals a misalignment between their objectives. Accurate segmentation as measured by the intersection over union (IoU) does not necessarily imply accurate object separation, and vice versa. For accurate detection and classification, it would be sufficient to only detect the interior of an object as long as the segmented area allows to correctly classify the type. If the roofs of two buildings are (almost) touching, then some segmentation may have a high IoU but could make it difficult to separate buildings for counting. This is also a common issue in other applications, e.g., when studying cells in medical images (Ronneberger et al., 2015) or trees from satellite images (Brandt et al., 2020; Mugabowindekwe et al., 2022)).

We benchmark three conceptually different state-of-the-art approaches on our multi-task dataset. First, we evaluate YOLOv8 (Jocher et al., 2023) developed for object detection, classification, and instance segmentation. Second, we build a segmentation model based on DINOv2 (Oquab et al., 2024), a state-of-the-art pretrained vision transformer. Lastly, we evaluate U-Net (Ronneberger et al., 2015) a fully-convolutional encoder-decoder architecture, designed for semantic segmentation. To address the potential conflicts between pixel-level segmentation and correct object separation as outlined above, we propose a simple approach based on the recent work by Cheng et al. (2024), which we refer to as the Deep Ordinal Watershed (DOW) method. We extend both U-Net and DINOv2 to produce an additional output map that predicts the interior of objects. While the original exterior segmentation map maximizes the IoU, we show that the interior map supports object separation.

The main contributions of our work are the following:

1. We provide the Nacala-Roof-Material dataset containing drone imagery from peri-urban and rural areas in a sub-Saharan African region. The dataset contains accurate segmentation labels for buildings, categorized into five roof types.
2. Based on the dataset, we define a multi-task machine learning benchmark for binary and multi-class object detection and semantic segmentation. We implemented and benchmarked different carefully adopted baseline methods, reflecting three different approaches to address these tasks.
3. We propose a general and simple approach to extend models for semantic segmentation to yield good segmentation *and* object separation results.

The data and code for reproducing the experiments are available through this anonymous url: `https://osf.io/us628/?view_only=3c25a48d420f4ec7a43cb76e66e92b26`. A project page, with link to data and code, will be setup upon acceptance.

The next section presents the Nacala-Roof-Material data, provides some background about roof types and risk of vector-borne diseases, and briefly discussed related datasets. Section 3 describes the deep learning models we evaluated with an emphasis on deep watershed methods. Experimental results are presented in Section 4 before we conclude.

## 2 NACALA-ROOF-MATERIAL DATA

### 2.1 BACKGROUND: HOUSING CONDITIONS AND RISK OF MOSQUITO-BORNE DISEASES

In sub-Saharan Africa, housing conditions, health outcomes, and socioeconomic status of the residents are interrelated (Gram-Hansen et al., 2019; Degarege et al., 2019; Tusting et al., 2020). As poverty is widespread, diseases are more prevalent, and data are scarce in this region, automatic profiling of housing conditions based on analysis of satellite imagery holds the potential to estimate the socioeconomic status of the inhabitants and assess the risk of disease. This may in turn support targeted public health interventions.

Mosquitoes are vectors for diseases such as malaria, dengue, Zika, West Nile fever, Chikungunya, and Yellow fever. In 2022, more than $600\,000$ deaths occurred due to malaria globally and out of the approximately 249 million documented cases, around 233 million occurred within the WHO African

Region, accounting for roughly around 94% of the total documented cases. The economic impact of malaria in Sub-Saharan Africa not only impedes progress towards achieving Sustainable Development Goal 3 (Good Health and Well-being) but also undermines efforts to attain SDG 1 (No Poverty) and SDG 8 (Decent Work and Economic Growth) by compromising economic productivity. Extreme weather conditions caused by climate change will likely exacerbate problems with mosquito-borne diseases in sub-Saharan Africa, as floods are expected to increase in frequency and have been linked to outbreaks of malaria in Africa (Githeko et al., 2000).

Low-quality housing increases the risk of transmission of diseases by mosquitoes, as sub-standard houses have more mosquito entry points and thereby increase human exposure to infection in the home (Tusting et al., 2015; Dlamini et al., 2017). Mosquito survival is lower in metal-roof houses compared to thatched-roof houses due to higher daytime temperatures (Tusting et al., 2015). Most malaria transmissions in sub-Saharan Africa occur indoors at night, and poor climatic performance of housing has been linked to increased malaria risk (Jatta et al., 2018). This is because elevated indoor temperatures can cause discomfort for inhabitants, which may result in decreased use of mosquito nets during the night. Roof materials, geometry, and conditions are critical for indoor climate, as roofs comprise the primary surface exposed to the sun. Automatic classification of roof characteristics thus holds potential for informing risk assessment of malaria and support targeted interventions.

## 2.2 THE NACALA-ROOF-MATERIAL DATASET

We gathered drone imagery of the Nacala region in Mozambique. The burden of malaria in Mozambique is approximately 10-fold the world average (number of documented cases compared to the total population, Venkatesan, 2024). The data covers three informal settlements of Nacala, a city of 350 000 inhabitants on the northern coast of Mozambique. Aerial imagery was collected using a DJI Phantom 4 Pro drone and processed using AgiSoft Metashape software. The flight height was 120 m, the total flight duration was 504 minutes (the drone flight protocols are available in the supplementary material). All data was recorded between October and December 2021, under a development project led by an NGO and supported by the Nacala Municipal Council.

The total number of buildings in the study areas is 17 954. We distinguished five major types of roof materials in Nacala, namely metal sheet, thatch, asbestos, concrete, and no-roof, and their counts are 9776, 6428, 566, 174, and 1010, respectively. The region is mostly dominated by metal sheets and thatch roofs.

From the three informal settlements, see Figure 1, the first two areas were split into training $\mathcal{D}_{train}$, validation $\mathcal{D}_{val}$, and test $\mathcal{D}_{test}$ using stratified sampling. We created a square grid of 225 meters and counted the roof types in these cells. Then we partitioned the cells into three sets based on the class counts to achieve a similar class distribution in each set, where we prioritized the distribution of minority classes (i.e., concrete and asbestos). We defined that a building only belongs to a specific grid cell if its centroid falls into the cell. If a building area falls into two grid cells and those two cells belong to two different sets (e.g., training and test set), we choose to have data pixels in the set where the centroid of the building is placed. The remaining part of the building in the other set was masked to avoid data leaking between sets.

Although objects in training, validation, and test sets are from different cells, they stem from the same two areas. To evaluate the generalization to a new area without adjacent training data, we hold out the third settlement as a second test set referred to as $\mathcal{D}_{ext}$.

## 2.3 ANNOTATION PROCESS AND QUALITY

Building boundary and roof type annotations were collected and corrected in three steps. First, local university students from the Nacala region and members of the NGO mapping community manually traced building boundaries, and collected roof types on-site, as part of a wider survey campaign. Local mapping teams used field papers and GPS tracking apps on smartphones. Secondly, the field data was then digitized, corroborated by observation of the drone orthomosaics. Finally, all building boundaries and roof types were verified by new observation of the drone orthomosaics conducted by the authors, and corrections made whenever necessary (almost all of the annotations were manually adjusted slightly in this second step, missing annotations were added). The authors created a grid over all annotations and verified all buildings in each grid cell to ensure double-checking every building.

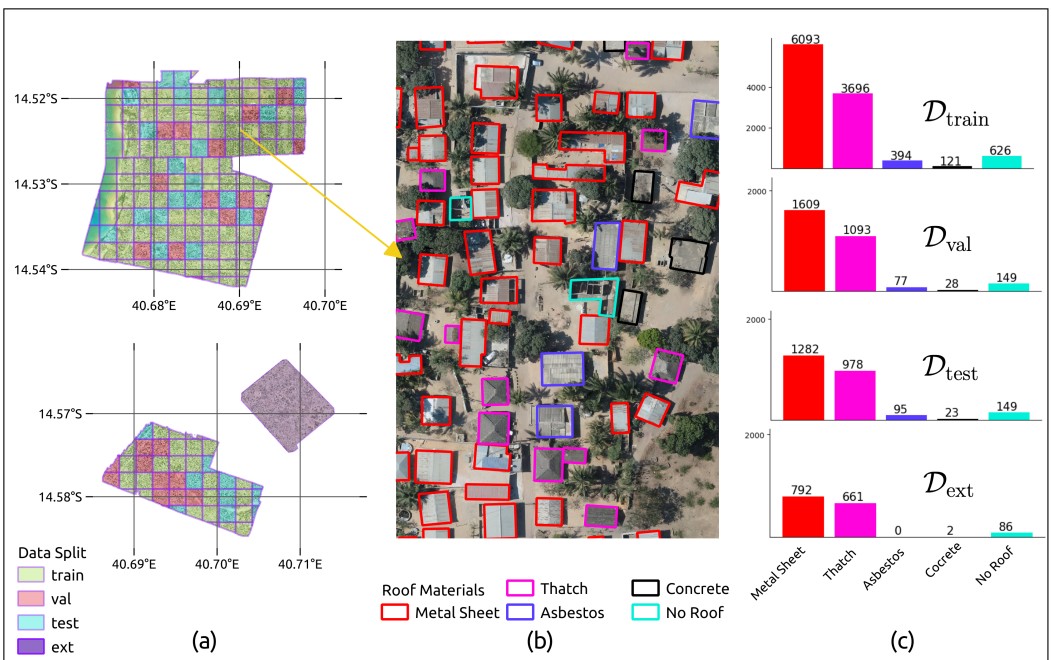

Figure 1: (a) Visualisation of the training, validation and test sets with reference to longitude and latitude; (b) Drone imagery with labels; (c) Instance counts for each class in all sets.

For the building boundaries, the quality can be accurately measured by direct observation of the drone orthomosaics. In rare cases, under very specific lighting and imagery conditions, some uncertainties can arise between two similarly looking roof types, for instance asbestos and concrete. These cases are, however, rare and do not compromise the overall quality of the annotated data. The three-steps process – having at least two people independently looking at the images and the labeling – ensured a high label quality. An estimated 120 person-hours were required for the first of these steps and around 40 person-hours for the second.

## 2.4 RELATED DATASETS

The project "Mapping Informal Settlements in Developing Countries using Machine Learning with Noisy Annotations and Multi-resolution Multi-spectral Data" (Helber et al., 2018; Gram-Hansen et al., 2019) is most closely related to our work. They used freely available 10m/pixel resolution imagery from the Sentinel-2 satellite and obtained labels for three roof types (metal, shingles, thatch) from geo-located survey data provided by Afrobarometer[1]. These labels are very noisy in space and time. The labels are often not aligned with buildings because the geo-located coordinates were distorted for privacy reasons. Furthermore, the survey questions and satellite image observations may not be aligned in time. While the low spatial resolution of the Sentinel-2 imagery might allow to cover large geographic regions, it makes roof type classification challenging (Helber et al., 2018).

There are many datasets that contain remote sensing imagery with building labels, which, however, typically do not distinguish roof types. In particular, *Open Buildings* is a freely available continental-scale building dataset covering the whole of Africa (Sirko et al., 2021). In comparison, Nacala-Roof-Material is much more focused, providing significantly higher resolution images, more accurate delineations, and in particular roof type classifications.

Alidoost and Arefi (2018) distinguish between roof types in aerial images. However, they map a rather high-income town in Germany, where they distinguish between three roof shapes common in that region (flat, gable, and hip). Another dataset for classifying roof geometry is provided by Persello et al. (2023), who distinguish 12 fine-grained details of roof geometry.

---

[1] www.afrobarometer.org

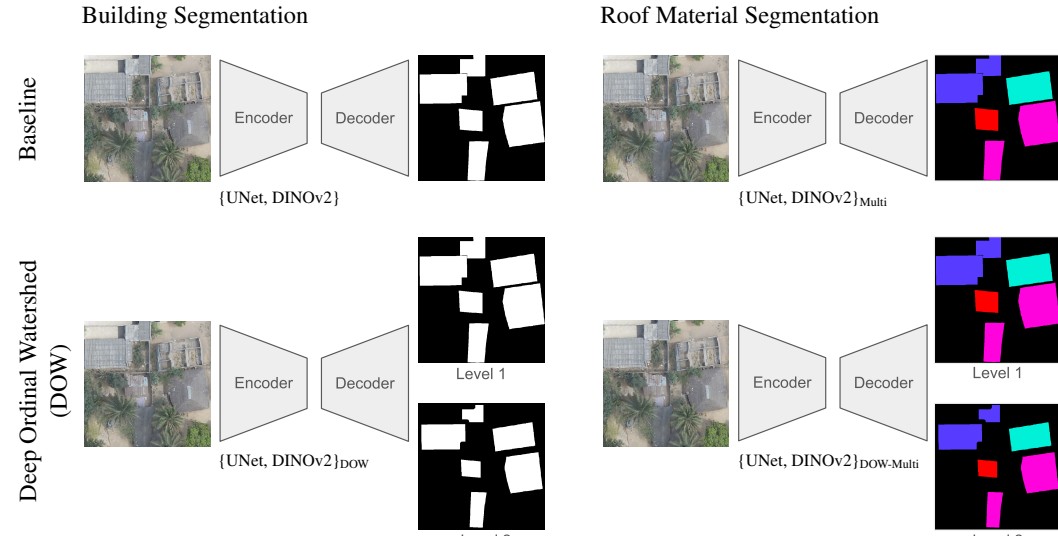

Figure 2: Baseline (top) and DOW (bottom) variants of our systems using either ResNet34 (in the case of the U-Net architectures) or DINOv2 as encoders. When using DOW, The watershed algorithm takes two segmentation masks as input, the predicted objects (level 1) and their interiors (level 2). In the two-stage approach, the classifier shown in Figure 4 is using the binary building segmentation (left). In the end-to-end setting, the roof material is predicted directly with a multi-class segmentation approach (right).

## 3 BENCHMARKED METHODS

This section presents the approaches we benchmarked on the Nacala-Roof-Material dataset. The goal is to accurately segment the buildings (as assessed by metrics based on the IoU), separate individual buildings, and classify the roof materials. As baselines, we considered U-Net (Ronneberger et al., 2015), YOLOv8 (Jocher et al., 2023), and a model performing segmentation based on DINOv2 (Oquab et al., 2024). Furthermore, we extend the U-Net and the DINOv2 based systems with the deep ordinal watershed method recently proposed by Cheng et al. (2024). These approaches are compared in two settings. In the *two-stage* setting, we first solved the building segmentation and separation tasks and afterwards classified the roof material for each detected building. In the *end-to-end* setting, segmentation and classification were done in parallel.

### 3.1 BASELINE MODELS

**U-Net.** The U-Net is arguably the most common architecture for semantic segmentation (Ronneberger et al., 2015). We utilized a ResNet34 (He et al., 2016) encoder pretrained on ImageNet and a decoder similar to the original U-Net, except that we used nearest-neighbor upsampling instead of transposed convolutions (Odena et al., 2016), see Figure A.6 in the Appendix.

To identify individual instances in the semantic segmentation output map, the connected components in the map were determined (Brandt et al., 2020). To better separate individual buildings, we used a pixel-wise weight map during training that puts more emphasis on the space between buildings as already suggested by Ronneberger et al. (see Appendix A.1 for details) and commonly used in remote sensing (e.g. Brandt et al., 2020). However, this is not sufficient to separate buildings that are very close to each other or touch each other. Thus, we modified the target segmentation masks during training: Some border pixels were relabeled as background to ensure that there is a minimum gap of $n_{\text{gap}} = 7$ pixels between roofs. This modification of the target masks was only applied during training, before computing the weight map but not when calculating any performance metrics.

**YOLOv8.** We trained YOLOv8 (Jocher et al., 2023), which is among the state-of-the-art methods for instance segmentation. We fine-tuned a model pretrained on the COCO dataset. While the original

YOLO architecture was designed for object detection, YOLOv8 allows for instance segmentation by integrating concepts from YOLACT (Bolya et al., 2019).

**DINOv2.** We benchmarked an approach based on DINOv2 (Oquab et al., 2024), a state-of-the-art pretrained vision transformer. It uses the DINOv2 *Base* model as an encoder, which is extended by a convolutional decoder. The DINOv2 output, a patch embedding with the shape of $\mathbb{R}^{1024 \times 768}$, is reshaped into feature maps of size $\mathbb{R}^{32 \times 32 \times 768}$. Then convolutional and linear upsampling layers are used on top of these feature maps as a decoder (see Appendix A.3). We used the same loss function, weighting function, training label adjustment, and training strategy as for U-Net. We froze the encoder weights and only the convolutional decoder was trained.

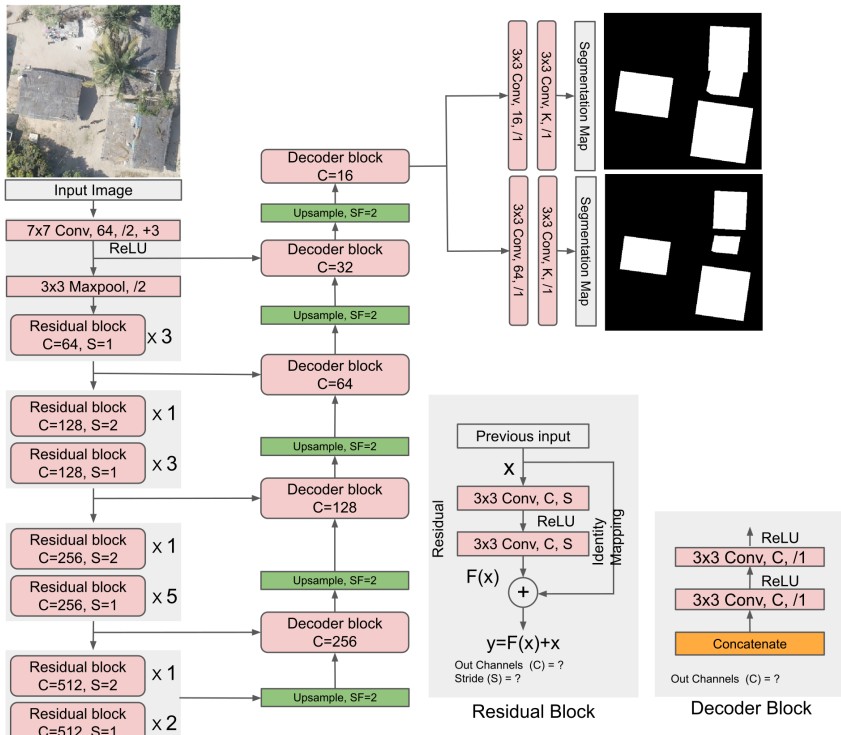

Figure 3: The U-Net$_{\text{DOW}}$ architecture creates two output maps that segment objects and their interiors, respectively. The architecture differs from the baseline U-Net only in the definition of their output heads.

## 3.2 DEEP ORDINAL WATERSHED

U-Nets and the DINOv2 based method described above try to classify each pixel as accurately as possible. However, for proper separation of objects it is sufficient – and typically preferable – if only the interior of an object is segmented. If the border of a building can be classified as background, even touching buildings can be separated. This reasoning leads to the deep ordinal watershed (DOW) model introduced by Cheng et al. (2024).

In the watershed approach, each pixel is assigned a height and the image is viewed as a topological map (Soille and Ansoult, 1990). A DOW architecture does not only predict a single segmentation mask but $n_{\text{lev}}$ feature maps for $n_{\text{lev}} + 1$ discrete height levels, $\{0, 1, \ldots, n_{\text{lev}}\}$, where $0$ corresponds to the highest and $n_{\text{lev}}$ to the lowest elevation. Background pixels are assumed to have level $0$. The Euclidean distance transformation is computed for each object, and the distances are discretized into the remaining $n_{\text{lev}}$ height levels. Target feature map $m \in \{1, \ldots, n_{\text{lev}}\}$ marks all pixel with a distance level of $m$ or higher. That is, the objects in the target feature maps get smaller with increasing $m$ (if $n_{\text{lev}} = 1$ we recover the standard U-Net). Learning the discrete height levels of pixels this way solves an ordinal regression task (Frank and Hall, 2001; Cheng et al., 2008). Given the pixel heights, the

watershed algorithm can be applied as a post-processing step for instance segmentation (Soille and Ansoult, 1990). Local minima in the elevation map define basins, each of which defines a distinct object. Adopting a flooding metaphor, the watershed algorithm now floods the basins until basins attributed to different starting points meet on watershed lines. Pixels attributed to the same basin belong to the same object.

Cheng et al. (2024) employ a DOW U-Net for individual tree segmentation, however, without a comparison with a standard U-Net or exploring different numbers of levels. For our task, we hypothesize that a minimal number of $n_{\text{lev}} = 2$ different non-background heights is sufficient. In this setting, the system outputs two masks representing the full object and its interior, respectively. Let $n_{\text{pix}}$ denote the difference in distance between two levels. The smallest building in our dataset has size $1.463\,\text{m}^2$. Thus, for the given image resolution, the number of pixels per side is approximately $\sqrt{1.46}/0.044$. This suggests to define the levels such that $n_{\text{pix}} < 13$, and we picked $n_{\text{pix}} = 10$.

We empirically evaluated DOW variants of both our U-Net and DINOv2 based systems, see Figure 2. We describe the U-Net extension in more detail in Appendix A.2, the DINOv2 based systems were modified analogously. The DOW U-Net network architecture U-Net$_{\text{DOW}}$ used in our study is illustrated in Figure 3. For a comparison with a DOW U-Net with $n_{\text{lev}} = 6$ we refer to Appendix A.2 and Appendix B.

Although the approaches are related, we would like to stress the DOW method is conceptually different from *deep level sets*, where deep neural networks learn a (continuous) level set function, the zero-set of which defines object boundaries (Hu et al., 2017; Hatamizadeh et al., 2020), as well as from predicting interior and border of an object as, for instance, done by Girard et al. (2021).

### 3.3 Two-stage vs. End-to-end

All the neural network architectures described above can directly classify the roof types of detected buildings by predicting multi-class segmentation masks. However, encouraged by good classification results using DINOv2 features, we also studied an alternative two-stage approach: First we segmented and separated the buildings using the algorithms described above ignoring the roof material information. That is, we reduced the multi-class problem to a binary task. After that, we predicted the roof material of each detected building. We used DINOv2 to processes a $448 \times 448$ patch centered around each building, see Figure 4. The output of DINOv2, a patch embedding with the shape of $\mathbb{R}^{1024 \times 768}$ was reshaped into feature maps of $\mathbb{R}^{32 \times 32 \times 768}$. These feature maps were then upsampled to the input patch size, masked with a target binary building mask, and average pooling was applied to obtain the final feature vector for the building. Standard machine learning classifiers were applied to this embedding to predict the roof material, where linear probing gave the best results (see Appendix B.2 for a comparison of different classifiers).

The two-stage methods are referred to as U-Net, DINOv2, U-Net$_{\text{DOW}}$, and DINOv2$_{\text{DOW}}$, and the corresponding end-to-end methods are denoted by U-Net$_{\text{Multi}}$, DINOv2$_{\text{DOW-Multi}}$, U-Net$_{\text{DOW-Multi}}$, and DINOv2$_{\text{DOW-Multi}}$, see Figure 2 for an overview.

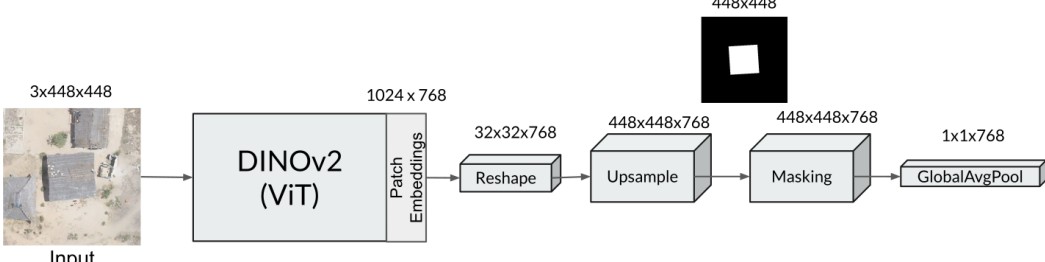

Figure 4: The architecture of the DINOv2 based roof material classifier used in the two-stage setting. A classifier (e.g., logistic regression) is applied to the resulting feature vector.

## 4 EXPERIMENTS AND RESULTS

### 4.1 EXPERIMENTAL SETUP

All models, except for YOLOv8 where we followed its original training protocol, were trained using cross-entropy loss with pixel-wise weighting. We employed the AdamW optimizer (Loshchilov and Hutter, 2019) with an initial learning rate of 0.0003. All models were trained for 300 epochs, utilizing a learning rate scheduler that decreased the learning rate by a factor of 10 every 50 epochs. The final weight configuration and hyperparameters for each model were selected based on the highest IoU score achieved on the validation dataset. The hyperparameters of the U-Net were chosen by observing results on the validation dataset in an iterative process. The high training speed of YOLOv8 allowed for more systematic model selection: We applied the genetic algorithm that comes as part of the YOLOv8 framework for hyperparameter optimization (Jocher et al., 2023). The input patch sizes for the U-Net variants, YOLOv8, and DINOv2 models were 512, 640, and 448, respectively.

### 4.2 EVALUATION METRICS

The semantic segmentation performance was evaluated by the IoU. We considered both the IoU of the binary building segmentation and the mean IoU for class-specific roof segmentation. The roof materials concrete and asbestos are very rare. While $\mathcal{D}_{\text{train}}$, $\mathcal{D}_{\text{val}}$, and $\mathcal{D}_{\text{test}}$ are stratified samples containing all classes, the spatially distinct data $\mathcal{D}_{\text{ext}}$ does not contain any example of the two roof types, see Figure 1. To allow for a better comparison between the two test sets and to see the effect of the rare classes on the macro-averaged mean IoU, we provide the mean IoU of the three main classes ($\text{mIoU}^3$) alongside with the mean IoU of all five classes ($\text{mIoU}^5$).

Instance segmentation was assessed using the $\text{AP}_{50}$ score, that is, the average precision evaluated at an IoU threshold of 0.5 (Everingham et al., 2010; Lin et al., 2014). We evaluated the AP for both the predictions of building instances and the predictions of multi-class roof type instances (i.e., in the latter case an object is only detected if the roof material is correctly identified). Similar to IoU, $\text{mAP}_{50}^3$ and $\text{mAP}_{50}^5$ denote the mean $\text{AP}_{50}$ over three and five classes. To estimate the average precision, a confidence score is required for each building segment. The confidence score of binary and multi-class segmentation models was obtained by interpreting the neural networks' outputs as probability distributions over classes and calculating the mean probability of belonging to the predicted class over all pixel within a predicted segment. The exception was YOLOv8, which provides its own confidence score. When a classifier using DINOv2 features was used on top of binary segmentation models, the confidence score was derived from the canonical probability score of the classifier. Additional metrics, $\text{AP}_{50\text{-}95}$ and $\text{TP}_{\text{s}}$, are shown in Appendix B. Information on the computer resources is provided in Appendix A.4.

### 4.3 RESULTS AND DISCUSSION

Our experimental results on $\mathcal{D}_{\text{test}}$ and $\mathcal{D}_{\text{ext}}$ are presented in Table 1, additional details can be found in Appendix B. All metrics on the test sets were computed on raw images instead of patches to avoid artifacts when splitting images. We report averages over five trials on the corresponding standard deviations. The methods reached $\text{AP}_{50}$ and IoU values on the spatially separated test set of up to 0.963 and 0.880, respectively. Thus the tasks can be solved with an accuracies high enough for subsequent analysis while still leaving room for improvement. Detecting thatch roofs is particularly relevant, as they are associated with an increased malaria risk (Tusting et al., 2019), and these roofs can be identified particularly well, see Table B.4 in the Appendix.

**Comparison of methods.**   When comparing the different approaches, we find that there is no method that was better than the others across all metrics. The U-Nets and YOLOv8 did well on their home grounds: YOLOv8 gave good object detection results (e.g., the best $\text{AP}_{50}$ scores), while the U-Nets performed well for semantic segmentation as measured by IoU. DINOv2 combined with a simple decoder was also competitive. Exemplary results are shown in Figure 5. As could be expected, classifying the minority roof types asbestos and especially concrete (which resembles concreted background areas) was most difficult, in particular for end-to-end YOLOv8, see Table B.4. YOLOv8 had the tendency to produce artefacts when applied to the larger images. This is one of the reasons for its lower IoU score.

Table 1: Benchmarking results on the Nacala-Roof-Material dataset. The table reports averages over five trials $\pm$ standard deviations. The upper five models were trained in the two-stage setting. The lower half of the models was trained in the end-to-end setting, where multi-class classification is performed together with the segmentation as indicated by the subscript *Multi*. Models that used the DOW extension are indicated by the subscript *DOW*. IoU and $AP_{50}$ were computed on the binary output, where the predictions of multi-class models were binarized. mIoU and $mAP_{50}$ are macro averages, the superscipts indicate whether the averaging was done over all five classes or over the three frequent roof types. Results for individual roof types can be found in Appendix B.

| Model Name | $\mathcal{D}_{test}$ | | | | | | $\mathcal{D}_{ext}$ | | | |
| | pixel level | | | object level | | | pixel level | | object level | |
| | IoU | $mIoU^3$ | $mIoU^5$ | $AP_{50}$ | $mAP_{50}^3$ | $mAP_{50}^5$ | IoU | $mIoU^3$ | $AP_{50}$ | $mAP_{50}^3$ |
|---|---|---|---|---|---|---|---|---|---|---|
| YOLOv8 | 0.866 $\pm$ 0.012 | 0.713 $\pm$ 0.019 | 0.568 $\pm$ 0.015 | **0.941** $\pm$ 0.003 | 0.815 $\pm$ 0.011 | 0.698 $\pm$ 0.018 | 0.896 $\pm$ 0.002 | 0.761 $\pm$ 0.006 | **0.963** $\pm$ 0.005 | 0.846 $\pm$ 0.008 |
| U-Net | **0.895** $\pm$ 0.003 | 0.757 $\pm$ 0.024 | 0.570 $\pm$ 0.016 | 0.910 $\pm$ 0.005 | 0.810 $\pm$ 0.008 | 0.688 $\pm$ 0.014 | 0.909 $\pm$ 0.001 | 0.748 $\pm$ 0.007 | 0.929 $\pm$ 0.000 | 0.787 $\pm$ 0.011 |
| U-Net$_{DOW}$ | **0.895** $\pm$ 0.002 | 0.775 $\pm$ 0.013 | 0.577 $\pm$ 0.009 | 0.935 $\pm$ 0.001 | 0.836 $\pm$ 0.005 | **0.730** $\pm$ 0.011 | **0.911** $\pm$ 0.002 | 0.764 $\pm$ 0.006 | 0.947 $\pm$ 0.004 | 0.812 $\pm$ 0.008 |
| DINOv2 | 0.880 $\pm$ 0.002 | 0.741 $\pm$ 0.002 | 0.549 $\pm$ 0.005 | 0.881 $\pm$ 0.004 | 0.783 $\pm$ 0.005 | 0.673 $\pm$ 0.011 | 0.904 $\pm$ 0.001 | 0.718 $\pm$ 0.015 | 0.922 $\pm$ 0.005 | 0.804 $\pm$ 0.008 |
| DINOv2$_{DOW}$ | 0.881 $\pm$ 0.001 | 0.761 $\pm$ 0.002 | 0.566 $\pm$ 0.004 | 0.931 $\pm$ 0.004 | 0.836 $\pm$ 0.003 | 0.718 $\pm$ 0.009 | 0.904 $\pm$ 0.001 | 0.767 $\pm$ 0.006 | 0.961 $\pm$ 0.005 | 0.861 $\pm$ 0.009 |
| YOLOv8$_{Multi}$ | 0.824 $\pm$ 0.023 | 0.708 $\pm$ 0.010 | 0.550 $\pm$ 0.017 | 0.910 $\pm$ 0.005 | 0.816 $\pm$ 0.009 | 0.597 $\pm$ 0.007 | 0.885 $\pm$ 0.002 | 0.785 $\pm$ 0.006 | 0.948 $\pm$ 0.003 | 0.849 $\pm$ 0.015 |
| U-Net$_{Multi}$ | 0.879 $\pm$ 0.012 | 0.783 $\pm$ 0.010 | 0.634 $\pm$ 0.024 | 0.924 $\pm$ 0.004 | **0.850** $\pm$ 0.011 | 0.716 $\pm$ 0.018 | 0.903 $\pm$ 0.002 | 0.805 $\pm$ 0.020 | 0.943 $\pm$ 0.010 | 0.844 $\pm$ 0.039 |
| U-Net$_{DOW-Multi}$ | 0.893 $\pm$ 0.002 | 0.779 $\pm$ 0.011 | 0.674 $\pm$ 0.041 | 0.933 $\pm$ 0.003 | 0.838 $\pm$ 0.005 | 0.710 $\pm$ 0.006 | 0.906 $\pm$ 0.002 | 0.798 $\pm$ 0.012 | 0.944 $\pm$ 0.005 | 0.830 $\pm$ 0.017 |
| DINOv2$_{Multi}$ | 0.879 $\pm$ 0.001 | 0.768 $\pm$ 0.005 | 0.694 $\pm$ 0.013 | 0.894 $\pm$ 0.004 | 0.810 $\pm$ 0.008 | 0.680 $\pm$ 0.019 | 0.899 $\pm$ 0.001 | 0.819 $\pm$ 0.006 | 0.940 $\pm$ 0.003 | 0.871 $\pm$ 0.015 |
| DINOv2$_{DOW-Multi}$ | 0.884 $\pm$ 0.002 | 0.783 $\pm$ 0.005 | 0.732 $\pm$ 0.008 | 0.918 $\pm$ 0.002 | 0.810 $\pm$ 0.009 | 0.701 $\pm$ 0.027 | 0.901 $\pm$ 0.001 | 0.823 $\pm$ 0.007 | 0.944 $\pm$ 0.005 | 0.843 $\pm$ 0.012 |

In general, the DOW extension improved both U-Nets and DINOv2 based architectures. Comparing DINOv2 with DINOv2$_{DOW}$ and U-Net with U-Net$_{DOW}$, the DOW variants were better in all ten performance indices (except for IoU on $\mathcal{D}_{test}$ where U-Net and U-Net$_{DOW}$ gave the same result). Comparing DINOv2$_{Multi}$ with DINOv2$_{DOW-Multi}$, the latter was better in all indicators except $mAP_{50}^3$ on $\mathcal{D}_{ext}$. Only for U-Net$_{DOW-Multi}$ the results were mixed, using DOW gave lower values for five indices and higher values for the other half. Overall, the DOW extension had a statistically significant positive effect on the object separation as intended. If we pool all 20 DOW trials and compare with the corresponding trials predicting a single mask, then the AP50 improved significantly (two-sided Wilcoxon rank sum test, $p < 0.001$) while the difference in IoU was not significant ($p > 0.05$).

**Computational requirements.** Since compute resources might be limited for researchers interested in this application, we analyze the runtime for deployment of these models (see Table A.2). While all methods run in a reasonable time, the end-to-end approach is faster than the two-stage approach, with U-Net$_{Multi}$ being the fastest and DINOv2$_{DOW}$ the slowest model. For example, mapping the entire city of Nacala (31910 ha), U-Net$_{Multi}$ would take approximately 6.36 GPU hours on a single AMD MI250X GPU with 64 GB VRAM.

**Dataset size.** We ran the experiments with an 80% stratified subset of the data, see Table B.8 in the supplementary material. The results changed only very slightly, indicating that our training dataset is representative and large enough for the defined task given the regional constraints.

**Limitations.** The Nacala-Roof-Material dataset is not a large-scale dataset by current standards and it is restricted to a single region. However, considering the proliferation of low-cost drone technologies, high-resolution geospatial surveying is becoming increasingly affordable and common in sub-Saharan Africa. Accordingly, similar but unlabelled data will likely become available in the coming years at large scale, which makes it important to develop methods to make good use of these

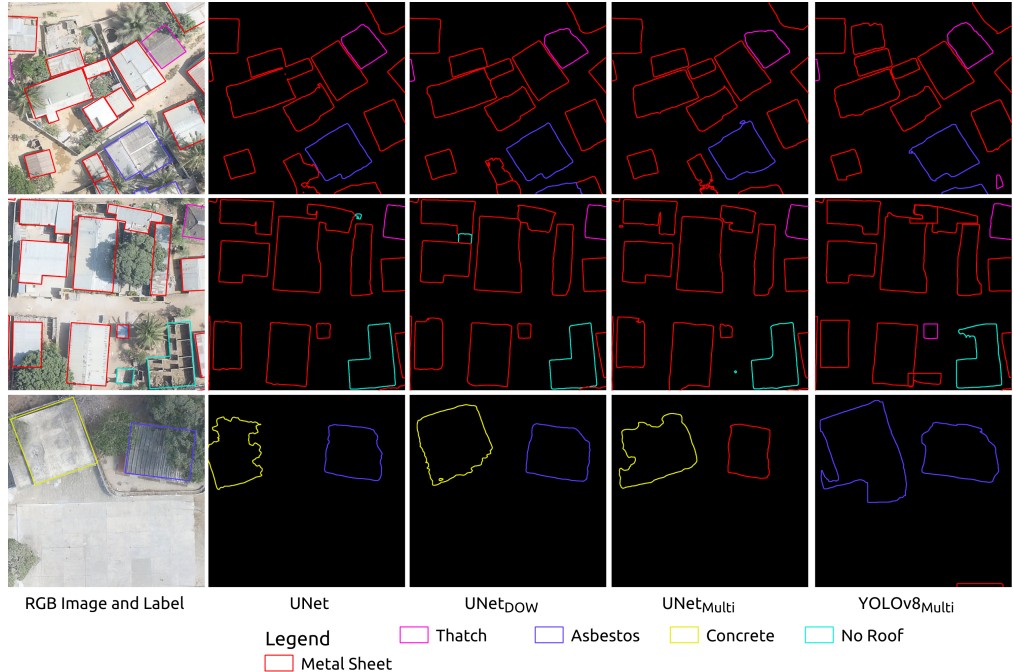

Figure 5: Exemplary predictions on $\mathcal{D}_{\text{test}}$ by different models. The predictions are polygonized and colored by class. The roof types with few training examples, asbestos and concrete, are particularly difficult, see bottom row.

data now. The Nacala-Roof-Material dataset covering informal settlements is a good example for the target areas of our risk disease monitoring and prevention research. In this context, Mozambique is particularly relevant because the country suffers from a high malaria incidence rate (Venkatesan, 2024). The second test set allows for testing generalization in an area geographically separated from the main training/test/validation data. In general, we would argue that there is a need for medium size benchmark datasets such as the Nacala-Roof-Material data to support equity in machine learning research, as we need benchmarks that can be utilized by researchers with limited compute resources.

## 5 Conclusions

The Nacala-Roof-Material dataset contains high-resolution drone imagery from informal settlements in Mozambique, where buildings and their roof material were carefully annotated. We curated the dataset as part of an intercontinental and interdisciplinary research project on risk assessment of mosquito-borne diseases, especially malaria, with the goal to predict risk maps and to develop and support measures for risk reduction. From a methodological perspective, the dataset defines a multi-task problem. We are interested in accurate semantic segmentation to determine the roof areas and also in identifying the individual buildings and classifying their roof types. Thus, the dataset adds to the landscape of computer vision benchmarks by providing a relevant resource for the development and evaluation of frameworks that strive at solving semantic segmentation as well as object detection and classification simultaneously with a high accuracy. For example, working on the Nacala-Roof-Material data has led us to the proposed deep ordinal watershed (DOW) approach, a reduced variant of the method described by Cheng et al. (2024). This variant method first segments objects along with their interiors into two elevation levels and then performs a watershed segmentation to separate objects. The DOW idea is applicable beyond the Nacala-Roof-Material data, on which it improved both the standard U-Net architectures as well as a system based on DINOv2 features for segmentation. Implementations of all algorithms will be publicly available together with the data. With the Nacala-Roof-Material dataset, we invite the machine learning community to develop new approaches for interpreting high-resolution drone images that can ultimately support risk assessments of vector-borne diseases.

# 6 REPRODUCIBILITY STATEMENT

The data and code for reproducing the experiments are available anonymously through `https://osf.io/us628/?view_only=3c25a48d420f4ec7a43cb76e66e92b26`. All material will be made freely available on an official project homepage upon acceptance.

# 7 ETHICS STATEMENT

We did not identify any ethical issues; we refer to the data sheet in Appendix C.

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

# A    DETAILS ON MODELS AND TRAINING PROCEDURE

## A.1    U-NET

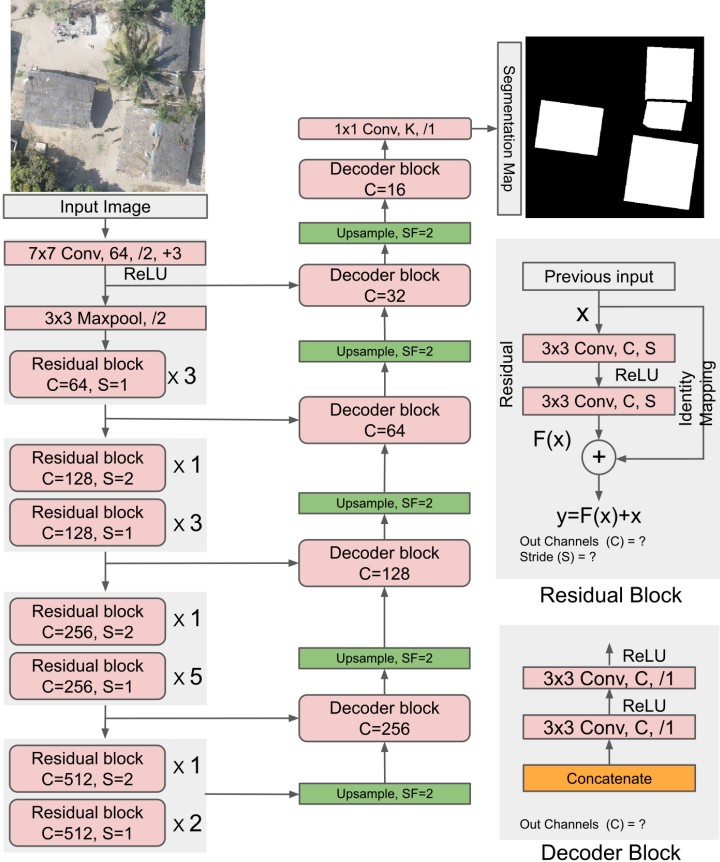

Figure A.6: Basic U-Net architecture

The basic U-Net architecture we used is shown in Figure A.6.

During training, the loss of each background pixel $x$ is multiplicatively weighted by $w(x)$ defined as

$$w(x) = w_0 \cdot \exp\left(-\frac{(d_1(x) + d_2(x))^2}{2\sigma^2}\right) \qquad (A.1)$$

following Ronneberger et al. (2015). Here, $d_1(x)$ denotes the distance to the border of the nearest segment, and $d_2(x)$ is the distance to the border of the second nearest segment. We set $w_0 = 10$ and $\sigma = 5$ according to Ronneberger et al. (2015).

During training, we modified the target masks to ensure that $d_1(x) + d_2(x) \geq n_{\text{gap}} = 7$ for each background pixel $x$ before we computed the weights $w(x)$.

## A.2    DEEP ORDINAL WATERSHED U-NETS

We considered a stripped down version of the DOW U-Net proposed by Cheng et al. (2024) and set the number of elevation levels to $n_{\text{lev}} = 2$. The architecture of the resulting DOW network is depicted in main text Figure 3, which extends the basic U-Net architecture shown in Figure A.6. In contrast to the original U-Net, the DOW model has two heads. One is predicting an object's area, while the other predicts its interior. The interior is defined by removing pixels within a 10-pixel distance from the border of the building segment. Each head comprises a convolutional layer, batch normalization, ReLU activation, and finally a pointwise convolutional layer with outputs equal to the number of

classes. While the first head had filters of size $3 \times 3$ in its first convolutional layer, the second head for the interior used 64 filters. The class label of an object was derived from the second head. If no interior was predicted, which can happen in the case of small objects, the output from the first head defined the class.

We compared this DOW variant, referred to as U-Net$_{\text{DOW}}$, to the original DOW with several elevation levels, in which the levels are added to the standard U-Net architecture (Figure A.6) simply by increasing the number of output masks. We considered $n_{\text{lev}} = 6$ discrete height levels and accordingly refer to the model as U-Net$_{\text{DOW-6}}$. The pixel margin $n_{\text{pix}}$ for each height level was determined experimentally by testing $n_{\text{pix}} \in \{1, 3, 5, 7, 9, 11, 13, 15\}$ on validation data, leading to $n_{\text{pix}} = 5$ for U-Net$_{\text{DOW-6}}$. An experimental comparison of U-Net$_{\text{DOW}}$ and U-Net$_{\text{DOW-6}}$ can be found in the extended results in Section B in the appendix.

### A.3 SEGMENTATION AND CLASSIFICATION USING DINOv2

The segmentation architecture based on DINOv2 is illustrated in Figure A.7. We refer to it simply as DINOv2. From this architecture, we derived DINOv2$_{\text{DOW}}$ in the same way as we extended U-Net to U-Net$_{\text{DOW}}$ .

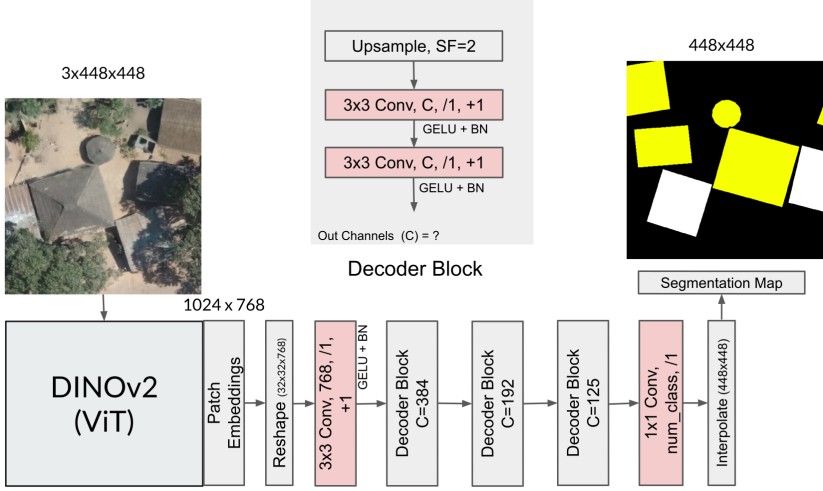

Figure A.7: DINOv2 architecture

### A.4 COMPUTE RESOURCES

All experiments were conducted on AMD MI250X GPUs provided by LUMI[2]. A total of 8550 GPU hours were used for the project, including preliminary experiments not included in the paper. The computation time for training semantic segmentation model was approximately 20 hours for 300 epochs when the entire data were loaded to GPU memory.

## B ADDITIONAL RESULTS

### B.1 DETAILED RESULTS FOR DIFFERENT ROOF MATERIALS

Additional results on $\mathcal{D}_{\text{test}}$ are presented in Table B.3 and Table B.4. The tables report the IoU scores for the individual roof material classes. They also show the true positive rates TP$_{\text{s}}$ in addition to the AP$_{50}$ the AP$_{50\text{-}95}$. The AP$_{50\text{-}95}$ is defined as the mean AP over IoU thresholds from $50\,\%$ to $95\,\%$ with an interval of $5\,\%$. The mean of AP$_{50\text{-}95}$ over all classes is mAP$_{50\text{-}95}$. TP$_{\text{s}}$ are the number of segments that overlap with ground truth segments with a minimum IoU of $0.5$, we used this metric to assess the counting of buildings.

---

[2]https://lumi-supercomputer.eu

Table A.2: Computation time deploying the models using a single AMD MI250X GPU with 64 GB VRAM. The prediction time includes segmentation, polygonisation, and post processing in case of the DOW method. The DOW method takes longer since its post-processing step currently runs on the CPU. We used our research code without any additional optimization for speed. We used our research code without optimization for speed.

| Model | Prediction time on $\mathcal{D}_{test}$ (minutes) |
|---|---|
| Two-stage approach | |
| YOLOv8 | 3.84 |
| U-Net | 3.53 |
| U-Net$_{DOW}$ | 5.37 |
| DINOv2 | 6.81 |
| DINOv2$_{DOW}$ | 11.10 |
| End-to-end approach | |
| YOLOv8$_{Multi}$ | 1.59 |
| U-Net$_{Multi}$ | 1.52 |
| U-Net$_{DOW-Multi}$ | 4.62 |
| DINOv2$_{Multi}$ | 5.36 |
| DINOv2$_{DOW-Multi}$ | 6.52 |

Beyond the performance metrics already discussed, we have included the results for U-Net$_{DOW-6}$ as described in Section A.2 in the appendix, showing that the two DOW architectures perform on par.

The corresponding results on $\mathcal{D}_{ext}$ are given in Table B.5 and Table B.6 The mean IoU in Table B.5, and mAP$_{50}$ and mAP$_{50-95}$ in Table B.6 estimated on only four classes as there are no asbestos roofs in $\mathcal{D}_{ext}$. Also, there are only two buildings of concrete found in $\mathcal{D}_{ext}$ and these two buildings were not identified from any of the experimental models, so results for the concrete class were not added to both tables.

### B.2 PERFORMANCE OF DIFFERENT CLASSIFIERS

In the two-stage approach, we used a classifier based on DINOv2 features, as described in Section 3.3 and illustrated in Figure 4. The input representation was fixed and was processed by standard classification algorithms. We compared linear probing based on logistic regression with $L_2$-regularization and k-nearest neighbours (kNN) classification trained on our data. For evaluating the classifiers and tuning their hyperparameters, we combined the training and validation data and performed 10-fold cross-validation (CV) with F1-score as performance metric. The best CV results gave logistic regression with $L_2$-regularization, and this model was used for all subsequent two-stage experiments, see Table B.2.

We also performed an ablation study to show the importance of the masking and the upsampling in our architecture shown in Figure 4. The results are also depicted in Table B.2. When we omitted the masking and considered all features, the results got considerably worse. If we omitted the upsampling of the DINOv2 output and downsampled the masks instead, the performance also slightly dropped.

Table B.3: Pixel-level accuracies on $\mathcal{D}_{\text{test}}$. IoU refers to the IoU computed on the binary outputs, where the predictions of multi-class models were binarized. mIoU[5] refers to the macro average of the IoUs for the individual classes. The subscript *Multi* indicates the end-to-end setting.

| | IoU-Score of each class | | | | | | |
|---|---|---|---|---|---|---|---|
| Model Name | Metal Sheet | Thatch | Asbestos | Concrete | No Roof | mIoU[5] | IoU |
| YOLOv8 | 0.807 ± 0.003 | 0.852 ± 0.038 | 0.450 ± 0.023 | 0.250 ± 0.027 | 0.480 ± 0.034 | 0.568 ± 0.015 | 0.866 ± 0.012 |
| DINOv2 | 0.796 ± 0.006 | 0.855 ± 0.003 | 0.335 ± 0.019 | 0.189 ± 0.017 | 0.571 ± 0.005 | 0.549 ± 0.005 | 0.880 ± 0.002 |
| DINOv2$_{\text{DOW}}$ | 0.814 ± 0.002 | 0.868 ± 0.002 | 0.351 ± 0.006 | 0.195 ± 0.019 | 0.602 ± 0.004 | 0.566 ± 0.004 | 0.881 ± 0.001 |
| U-Net | 0.813 ± 0.009 | 0.881 ± 0.002 | 0.408 ± 0.012 | 0.171 ± 0.021 | 0.577 ± 0.073 | 0.570 ± 0.016 | **0.895** ± 0.003 |
| U-Net$_{\text{DOW}}$ | 0.824 ± 0.005 | 0.879 ± 0.010 | 0.384 ± 0.042 | 0.174 ± 0.010 | 0.623 ± 0.028 | 0.577 ± 0.009 | **0.895** ± 0.002 |
| U-Net$_{\text{DOW-6}}$ | 0.824 ± 0.006 | **0.887** ± 0.002 | 0.424 ± 0.055 | 0.160 ± 0.026 | 0.591 ± 0.057 | 0.577 ± 0.011 | 0.888 ± 0.009 |
| YOLOv8$_{\text{Multi}}$ | 0.750 ± 0.030 | 0.824 ± 0.004 | 0.405 ± 0.021 | 0.223 ± 0.059 | 0.549 ± 0.026 | 0.550 ± 0.017 | 0.824 ± 0.023 |
| DINOv2$_{\text{Multi}}$ | 0.824 ± 0.002 | 0.866 ± 0.002 | 0.491 ± 0.022 | 0.675 ± 0.044 | 0.614 ± 0.015 | 0.694 ± 0.013 | 0.879 ± 0.001 |
| DINOv2$_{\text{DOW-Multi}}$ | **0.840** ± 0.003 | 0.874 ± 0.002 | **0.545** ± 0.015 | **0.767** ± 0.021 | 0.634 ± 0.014 | **0.732** ± 0.008 | 0.884 ± 0.002 |
| U-Net$_{\text{Multi}}$ | 0.819 ± 0.012 | 0.880 ± 0.004 | 0.514 ± 0.025 | 0.306 ± 0.091 | **0.650** ± 0.029 | 0.634 ± 0.024 | 0.879 ± 0.012 |
| U-Net$_{\text{DOW-Multi}}$ | 0.830 ± 0.022 | 0.884 ± 0.003 | 0.502 ± 0.039 | 0.533 ± 0.194 | 0.623 ± 0.017 | 0.674 ± 0.041 | 0.893 ± 0.002 |

Table B.4: Object-level accuracy on $\mathcal{D}_{\text{test}}$. We report the AP for each roof type, and $mAP_{50}$ and $mAP_{50\text{-}95}$ are macro averages over the roof types. The rightmost three columns give the results when we discard the roof type information and just consider building detection. The $TP_s$ columns count true positives, where $TP_s$ are the number of objects that overlap with ground truth objects with a minimum IoU of 0.5. The total number of ground truth objects in the $\mathcal{D}_{\text{test}}$ is 2527.

| | AP$_{50}$ of each class | | | | | average over classes | | | ignoring roof type | | |
|---|---|---|---|---|---|---|---|---|---|---|---|
| Model Name | Metal Sheet | Thatch | Asbestos | Concrete | No Roof | mAP$_{50}$ | mAP$_{50\text{-}95}$ | TP$_s$ | AP$_{50}$ | AP$_{50\text{-}95}$ | TP$_s$ |
| YOLOv8 | 0.841 ± 0.003 | 0.945 ± 0.008 | 0.505 ± 0.032 | 0.542 ± 0.055 | 0.661 ± 0.026 | 0.698 ± 0.018 | 0.548 ± 0.010 | 2262.2 ± 7.386 | **0.941** ± 0.003 | 0.798 ± 0.002 | **2405.0** ± 5.514 |
| DINOv2 | 0.799 ± 0.006 | 0.892 ± 0.004 | 0.455 ± 0.023 | 0.565 ± 0.034 | 0.657 ± 0.016 | 0.673 ± 0.011 | 0.526 ± 0.005 | 2131.0 ± 6.841 | 0.881 ± 0.004 | 0.732 ± 0.004 | 2257.4 ± 8.089 |
| DINOv2$_{\text{DOW}}$ | 0.849 ± 0.005 | 0.944 ± 0.004 | 0.478 ± 0.005 | **0.601** ± 0.036 | 0.717 ± 0.009 | 0.718 ± 0.009 | 0.568 ± 0.004 | 2236.2 ± 5.636 | 0.931 ± 0.004 | 0.780 ± 0.002 | 2375.8 ± 5.115 |
| U-Net | 0.826 ± 0.005 | 0.924 ± 0.006 | 0.499 ± 0.016 | 0.511 ± 0.042 | 0.679 ± 0.015 | 0.688 ± 0.014 | 0.578 ± 0.014 | 2191.2 ± 11.25 | 0.910 ± 0.005 | 0.797 ± 0.003 | 2323.0 ± 6.033 |
| U-Net$_{\text{DOW}}$ | 0.855 ± 0.005 | **0.946** ± 0.005 | 0.545 ± 0.019 | 0.596 ± 0.049 | 0.707 ± 0.012 | **0.730** ± 0.011 | **0.614** ± 0.007 | 2249.4 ± 4.128 | 0.935 ± 0.001 | **0.819** ± 0.003 | 2383.6 ± 5.314 |
| U-Net$_{\text{DOW-6}}$ | 0.851 ± 0.006 | 0.943 ± 0.004 | **0.551** ± 0.011 | 0.587 ± 0.049 | 0.687 ± 0.022 | 0.724 ± 0.007 | 0.606 ± 0.005 | 2243.2 ± 3.487 | 0.929 ± 0.004 | 0.818 ± 0.002 | 2374.4 ± 5.783 |
| YOLOv8$_{\text{Multi}}$ | 0.849 ± 0.006 | 0.923 ± 0.007 | 0.467 ± 0.035 | 0.070 ± 0.027 | 0.676 ± 0.020 | 0.597 ± 0.003 | 0.481 ± 0.003 | 2195.6 ± 15.383 | 0.910 ± 0.005 | 0.751 ± 0.007 | 2328.2 ± 9.988 |
| DINOv2$_{\text{Multi}}$ | 0.866 ± 0.004 | 0.927 ± 0.006 | 0.445 ± 0.043 | 0.525 ± 0.076 | 0.637 ± 0.028 | 0.680 ± 0.019 | 0.509 ± 0.008 | 2230.8 ± 5.344 | 0.894 ± 0.004 | 0.732 ± 0.002 | 2305.6 ± 5.678 |
| DINOv2$_{\text{DOW-Multi}}$ | **0.885** ± 0.005 | 0.942 ± 0.006 | 0.542 ± 0.026 | 0.529 ± 0.122 | 0.605 ± 0.027 | 0.701 ± 0.027 | 0.557 ± 0.027 | 2270.2 ± 8.376 | 0.918 ± 0.002 | 0.769 ± 0.004 | 2360.8 ± 7.467 |
| U-Net$_{\text{Multi}}$ | 0.883 ± 0.009 | 0.940 ± 0.010 | 0.531 ± 0.022 | 0.498 ± 0.051 | **0.728** ± 0.040 | 0.716 ± 0.018 | 0.603 ± 0.011 | 2262.4 ± 6.119 | 0.924 ± 0.004 | 0.797 ± 0.007 | 2358.4 ± 9.091 |
| U-Net$_{\text{DOW-Multi}}$ | 0.894 ± 0.006 | 0.943 ± 0.000 | 0.516 ± 0.038 | 0.517 ± 0.038 | 0.678 ± 0.018 | 0.710 ± 0.006 | 0.606 ± 0.007 | **2275.2** ± 5.879 | 0.933 ± 0.003 | 0.813 ± 0.003 | 2382.0 ± 9.055 |

Table B.5: Pixel-level accuracies on $\mathcal{D}_{\text{ext}}$. IoU refers to the IoU computed on the binary outputs, where the predictions of multi-class models were binarized. mIoU[5] refers to the macro average of the IoUs for the individual classes. The subscript *Multi* indicates the end-to-end setting.

| Model Name | IoU-Score of each class | | | IoU (Mean) | IoU (Binary) |
|---|---|---|---|---|---|
| | Metal Sheet | Thatch | No Roof | | |
| YOLOv8 | 0.888 ± 0.003 | 0.879 ± 0.003 | 0.516 ± 0.011 | 0.761 ± 0.006 | 0.896 ± 0.002 |
| DINOv2 | 0.844 ± 0.012 | 0.854 ± 0.005 | 0.456 ± 0.037 | 0.718 ± 0.015 | 0.904 ± 0.001 |
| DINOv2$_{\text{DOW}}$ | 0.886 ± 0.004 | 0.875 ± 0.003 | 0.541 ± 0.013 | 0.767 ± 0.006 | 0.904 ± 0.001 |
| U-Net | 0.896 ± 0.005 | 0.883 ± 0.005 | 0.463 ± 0.017 | 0.748 ± 0.007 | 0.909 ± 0.001 |
| U-Net$_{\text{DOW}}$ | 0.905 ± 0.002 | **0.895** ± 0.003 | 0.493 ± 0.018 | 0.764 ± 0.006 | **0.911** ± 0.002 |
| U-Net$_{\text{DOW-6}}$ | 0.900 ± 0.008 | 0.889 ± 0.002 | 0.452 ± 0.031 | 0.747 ± 0.009 | 0.902 ± 0.003 |
| YOLOv8$_{\text{Multi}}$ | 0.890 ± 0.006 | 0.860 ± 0.006 | 0.606 ± 0.019 | 0.785 ± 0.006 | 0.885 ± 0.002 |
| DINOv2$_{\text{Multi}}$ | 0.907 ± 0.001 | 0.874 ± 0.004 | **0.676** ± 0.022 | 0.819 ± 0.006 | 0.899 ± 0.001 |
| DINOv2$_{\text{DOW-Multi}}$ | 0.912 ± 0.001 | 0.881 ± 0.003 | **0.676** ± 0.020 | **0.823** ± 0.007 | 0.901 ± 0.001 |
| U-Net$_{\text{Multi}}$ | **0.913** ± 0.005 | 0.884 ± 0.003 | 0.617 ± 0.061 | 0.805 ± 0.020 | 0.903 ± 0.002 |
| U-Net$_{\text{DOW-Multi}}$ | **0.913** ± 0.005 | 0.884 ± 0.003 | 0.617 ± 0.061 | 0.805 ± 0.020 | 0.903 ± 0.002 |

Table B.6: Object-level accuracies on $\mathcal{D}_{ext}$. We report the AP for each roof type, and $mAP_{50}$ and $mAP_{50\text{-}95}$ are macro averages over the classes. The rightmost three columns give the results when we discard the roof type information and just consider building detection. $TP_s$ are the number of objects that overlap with ground truth objects with a minimum IoU of $0.5$. The total number of ground truth objects in the $\mathcal{D}_{ext}$ is $1541$.

| Model Name | AP$_{50}$ of each class | | | Objects with Classes | | | Only Building Objects | | |
| --- | --- | --- | --- | --- | --- | --- | --- | --- | --- |
| | Metal Sheet | Thatch | No Roof | mAP$_{50}$ | mAP$_{50\text{-}95}$ | TP$_s$ | AP$_{50}$ | AP$_{50\text{-}95}$ | TP$_s$ |
| YOLOv8 | 0.928 ± 0.001 | **0.947** ± 0.000 | 0.661 ± 0.023 | 0.846 ± 0.008 | 0.428 ± 0.002 | 1447.2 ± 4.534 | **0.963** ± 0.005 | 0.838 ± 0.002 | 1493.8 ± 3.826 |
| DINOv2 | 0.896 ± 0.010 | 0.883 ± 0.012 | 0.634 ± 0.019 | 0.804 ± 0.008 | 0.390 ± 0.003 | 1382.8 ± 8.818 | 0.922 ± 0.005 | 0.785 ± 0.005 | 1434.2 ± 6.524 |
| DINOv2$_{DOW}$ | 0.932 ± 0.007 | 0.944 ± 0.005 | 0.709 ± 0.016 | 0.861 ± 0.009 | 0.426 ± 0.004 | 1444.8 ± 5.455 | 0.961 ± 0.005 | 0.830 ± 0.003 | **1494.4** ± 4.363 |
| U-Net | 0.915 ± 0.006 | 0.921 ± 0.006 | 0.520 ± 0.027 | 0.590 ± 0.006 | 0.407 ± 0.004 | 1399.4 ± 4.758 | 0.929 ± 0.000 | 0.836 ± 0.002 | 1438.4 ± 4.499 |
| U-Net$_{DOW}$ | 0.932 ± 0.003 | 0.946 ± 0.004 | 0.559 ± 0.027 | 0.812 ± 0.008 | 0.528 ± 0.003 | 1429.0 ± 6.229 | 0.947 ± 0.004 | **0.858** ± 0.004 | 1468.6 ± 6.499 |
| U-Net$_{DOW\text{-}6}$ | 0.935 ± 0.001 | 0.940 ± 0.004 | 0.509 ± 0.022 | 0.795 ± 0.008 | 0.518 ± 0.005 | 1421.0 ± 3.688 | 0.939 ± 0.000 | 0.851 ± 0.004 | 1458.8 ± 4.118 |
| YOLOv8$_{Multi}$ | 0.949 ± 0.004 | 0.934 ± 0.007 | 0.664 ± 0.044 | 0.849 ± 0.015 | 0.423 ± 0.008 | 1446.0 ± 4.899 | 0.948 ± 0.003 | 0.808 ± 0.005 | 1477.2 ± 3.655 |
| DINOv2$_{Multi}$ | 0.951 ± 0.004 | 0.929 ± 0.003 | **0.732** ± 0.043 | **0.871** ± 0.015 | 0.424 ± 0.007 | **1453.2** ± 5.154 | 0.940 ± 0.003 | 0.796 ± 0.003 | 1467.8 ± 5.154 |
| DINOv2$_{DOW\text{-}Multi}$ | 0.955 ± 0.002 | 0.942 ± 0.010 | 0.633 ± 0.028 | 0.843 ± 0.012 | 0.503 ± 0.038 | 1462.2 ± 5.192 | 0.944 ± 0.005 | 0.810 ± 0.002 | 1481.4 ± 4.499 |
| U-Net$_{Multi}$ | **0.956** ± 0.004 | 0.926 ± 0.008 | 0.651 ± 0.107 | 0.844 ± 0.039 | **0.548** ± 0.017 | 1439.0 ± 16.358 | 0.943 ± 0.010 | 0.838 ± 0.006 | 1463.6 ± 13.063 |
| U-Net$_{DOW\text{-}Multi}$ | 0.956 ± 0.004 | 0.926 ± 0.008 | 0.651 ± 0.107 | 0.844 ± 0.039 | 0.438 ± 0.017 | 1439.0 ± 16.358 | 0.943 ± 0.010 | 0.838 ± 0.006 | 1463.6 ± 13.063 |

Table B.7: Multi-class prediction using DOW-Multi models. There are several ways to derive multi-class predictions in the DOW models. The approach in the main part of this study derives class labels by taking the majority vote of all inner segmentations combined with the border pixels from the outer mask that do not overlap with the inner one. Alternatively, one could solely consider only the inner mask or outer mask, indicated by the suffixes *-inner* and *-outer*, respectively, in the subscripts of the model suffixes. This table adds the results for these alternative methods, reporting again averages over five trials ± standard deviations. The upper five models were added to compare DINOv2 models and the lower half of the models were added using U-Net-based methods.

| Model Name | $\mathcal{D}_{test}$ | | | | | | $\mathcal{D}_{ext}$ | | | |
| --- | --- | --- | --- | --- | --- | --- | --- | --- | --- | --- |
| | pixel level | | | object level | | | pixel level | | object level | |
| | IoU | mIoU$^3$ | mIoU$^5$ | AP$_{50}$ | mAP$_{50}^3$ | mAP$_{50}^5$ | IoU | mIoU$^3$ | AP$_{50}$ | mAP$_{50}^3$ |
| DINOv2$_{Multi}$ | 0.879 ± 0.001 | 0.768 ± 0.005 | 0.694 ± 0.013 | 0.894 ± 0.004 | 0.810 ± 0.008 | 0.680 ± 0.019 | 0.899 ± 0.001 | 0.819 ± 0.006 | 0.940 ± 0.003 | 0.871 ± 0.015 |
| DINOv2$_{DOW\text{-}Multi}$ | 0.884 ± 0.002 | **0.783** ± 0.005 | **0.732** ± 0.009 | 0.918 ± 0.002 | 0.810 ± 0.009 | 0.701 ± 0.027 | 0.901 ± 0.001 | 0.822 ± 0.007 | 0.944 ± 0.005 | 0.843 ± 0.012 |
| DINOv2$_{DOW\text{-}Multi\text{-}outer}$ | 0.884 ± 0.002 | 0.783 ± 0.005 | 0.732 ± 0.008 | 0.918 ± 0.002 | 0.810 ± 0.009 | 0.701 ± 0.027 | 0.901 ± 0.001 | 0.823 ± 0.007 | 0.944 ± 0.005 | 0.843 ± 0.012 |
| DINOv2$_{DOW\text{-}Multi\text{-}inner}$ | 0.884 ± 0.002 | 0.782 ± 0.005 | 0.730 ± 0.009 | 0.918 ± 0.002 | 0.810 ± 0.009 | 0.699 ± 0.027 | 0.901 ± 0.001 | 0.821 ± 0.007 | 0.944 ± 0.005 | 0.842 ± 0.012 |
| U-Net$_{Multi}$ | 0.879 ± 0.012 | 0.783 ± 0.010 | 0.634 ± 0.024 | 0.924 ± 0.004 | **0.850** ± 0.011 | 0.716 ± 0.018 | 0.903 ± 0.002 | 0.805 ± 0.020 | 0.943 ± 0.010 | 0.844 ± 0.039 |
| U-Net$_{DOW\text{-}Multi}$ | 0.893 ± 0.002 | 0.779 ± 0.011 | 0.674 ± 0.041 | 0.933 ± 0.003 | 0.838 ± 0.005 | 0.709 ± 0.006 | 0.906 ± 0.002 | 0.798 ± 0.012 | 0.944 ± 0.005 | 0.830 ± 0.017 |
| U-Net$_{DOW\text{-}Multi\text{-}outer}$ | 0.893 ± 0.002 | 0.779 ± 0.011 | 0.674 ± 0.041 | 0.933 ± 0.003 | 0.838 ± 0.005 | 0.710 ± 0.006 | 0.906 ± 0.002 | 0.798 ± 0.012 | 0.944 ± 0.005 | 0.830 ± 0.017 |
| U-Net$_{DOW\text{-}Multi\text{-}inner}$ | 0.893 ± 0.002 | 0.779 ± 0.011 | 0.673 ± 0.040 | 0.933 ± 0.003 | 0.838 ± 0.006 | 0.709 ± 0.006 | 0.906 ± 0.002 | 0.797 ± 0.012 | 0.944 ± 0.005 | 0.830 ± 0.017 |

Table B.8: We ran the experiments with an 80% stratified subset of the data. The results changed only very slightly, indicating that our training dataset is representative and large enough for the defined task.

| | $\mathcal{D}_{test}$ | | | | | | $\mathcal{D}_{ext}$ | | | |
| | pixel level | | | object level | | | pixel level | | object level | |
| Model Name | IoU | $mIoU^3$ | $mIoU^5$ | $AP_{50}$ | $mAP_{50}^3$ | $mAP_{50}^5$ | IoU | $mIoU^3$ | $AP_{50}$ | $mAP_{50}^3$ |
|---|---|---|---|---|---|---|---|---|---|---|
| YOLOv8 | 0.855 ± 0.020 | 0.715 ± 0.018 | 0.560 ± 0.014 | **0.937** ± 0.004 | 0.823 ± 0.008 | 0.679 ± 0.009 | 0.894 ± 0.001 | 0.773 ± 0.006 | 0.960 ± 0.005 | 0.854 ± 0.008 |
| U-Net | 0.893 ± 0.007 | 0.758 ± 0.021 | 0.571 ± 0.004 | 0.914 ± 0.004 | 0.824 ± 0.008 | 0.683 ± 0.009 | **0.912** ± 0.002 | 0.738 ± 0.011 | 0.941 ± 0.004 | 0.818 ± 0.017 |
| U-Net$_{DOW}$ | **0.896** ± 0.003 | **0.792** ± 0.006 | 0.582 ± 0.003 | 0.933 ± 0.001 | **0.846** ± 0.005 | 0.695 ± 0.011 | 0.911 ± 0.001 | 0.743 ± 0.011 | 0.953 ± 0.005 | 0.820 ± 0.014 |
| DINOv2 | 0.880 ± 0.002 | 0.721 ± 0.027 | 0.535 ± 0.010 | 0.883 ± 0.001 | 0.793 ± 0.008 | 0.657 ± 0.014 | 0.904 ± 0.001 | 0.729 ± 0.008 | 0.928 ± 0.005 | 0.825 ± 0.006 |
| DINOv2$_{DOW}$ | 0.879 ± 0.001 | 0.758 ± 0.006 | 0.554 ± 0.006 | 0.931 ± 0.004 | 0.841 ± 0.005 | 0.697 ± 0.007 | 0.903 ± 0.001 | 0.755 ± 0.007 | **0.961** ± 0.004 | 0.863 ± 0.006 |
| YOLOv8$_{Multi}$ | 0.806 ± 0.028 | 0.697 ± 0.018 | 0.536 ± 0.029 | 0.901 ± 0.002 | 0.798 ± 0.010 | 0.583 ± 0.009 | 0.881 ± 0.001 | 0.777 ± 0.007 | 0.942 ± 0.003 | 0.832 ± 0.004 |
| U-Net$_{Multi}$ | 0.886 ± 0.005 | 0.786 ± 0.008 | 0.666 ± 0.023 | 0.924 ± 0.004 | 0.839 ± 0.011 | **0.712** ± 0.023 | 0.906 ± 0.001 | 0.799 ± 0.011 | 0.948 ± 0.000 | 0.839 ± 0.012 |
| U-Net$_{DOW-Multi}$ | 0.887 ± 0.003 | 0.782 ± 0.009 | 0.670 ± 0.044 | 0.929 ± 0.005 | 0.837 ± 0.011 | 0.709 ± 0.013 | 0.905 ± 0.003 | 0.812 ± 0.012 | 0.951 ± 0.004 | 0.855 ± 0.020 |
| DINOv2$_{Multi}$ | 0.870 ± 0.004 | 0.771 ± 0.004 | 0.675 ± 0.004 | 0.901 ± 0.005 | 0.824 ± 0.013 | 0.690 ± 0.023 | 0.895 ± 0.003 | 0.809 ± 0.013 | 0.945 ± 0.007 | **0.870** ± 0.022 |
| DINOv2$_{DOW-Multi}$ | 0.880 ± 0.001 | 0.785 ± 0.004 | **0.738** ± 0.008 | 0.911 ± 0.008 | 0.823 ± 0.008 | 0.703 ± 0.036 | 0.899 ± 0.002 | **0.819** ± 0.003 | 0.955 ± 0.004 | 0.868 ± 0.009 |

Table B.9: Cross-validation accuracies on combined training and validation data of k-nearest neighbour classification (kNN) and logistic regression applied to the DINOv2 features. The baseline is the architecture depicted in Figure 4, *w/o mask* refers to omitting the masking and averaging the DINOv2 features across the whole input patch, and *w/o upsampling* did not upsample the DINOv2 features but downsampled the building mask instead.

| | F1-Score | | | | | |
| | baseline | | w/o mask | | w/o upsampling | |
| Classifier | Mean | Std | Mean | Std | Mean | Std |
|---|---|---|---|---|---|---|
| Logistic Regression | **0.770** | 0.063 | 0.573 | 0.077 | 0.768 | 0.067 |
| kNN | 0.734 | 0.045 | 0.389 | 0.029 | 0.733 | 0.051 |

# C  DATASHEET

## C.1  MOTIVATION

### C.1.1  FOR WHAT PURPOSE WAS THE DATASET CREATED? WAS THERE A SPECIFIC TASK IN MIND? WAS THERE A SPECIFIC GAP THAT NEEDED TO BE FILLED?

The dataset was created to support research on multi-task computer vision problems and to support mosquito-borne disease risk assessment in African cities. The list of tasks include classification, semantic segmentation, and instance segmentation of roofs and their material. While these tasks are closely related, each serves a different purpose and accurate segmentation of objects need not imply accurate object separation, and vice versa. The dataset is ideal for bench-marking methods for the above mentioned tasks.

### C.1.2 WHO CREATED THE DATASET (E.G., WHICH TEAM, RESEARCH GROUP) AND ON BEHALF OF WHICH ENTITY (E.G., COMPANY, INSTITUTION, ORGANIZATION)?

The dataset was created by authors. The drone imagery and building footprints were captured by an NGO. The imagery and the building footprints were fused, re-registered, cleaned, verified, and split into given datasets by the authors.

### C.1.3 WHO FUNDED THE CREATION OF THE DATASET? IF THERE IS AN ASSOCIATED GRANT, PLEASE PROVIDE THE NAME OF THE GRANTOR AND THE GRANT NAME AND NUMBER

We will provide the project and grant details later.

### C.1.4 ANY OTHER COMMENT?

None.

## C.2 COMPOSITION

### C.2.1 WHAT DO THE INSTANCES THAT COMPRISE THE DATASET REPRESENT (E.G., DOCUMENTS, PHOTOS, PEOPLE, COUNTRIES)? ARE THERE MULTIPLE TYPES OF INSTANCES (E.G., MOVIES, USERS, AND RATINGS; PEOPLE AND INTERACTIONS BETWEEN THEM; NODES AND EDGES)? PLEASE PROVIDE A DESCRIPTION

The dataset comprises of very-high resolution orthophotos captured through a drone and expert drawn polygons for all buildings with annotation of their roof material. The dataset covers three informal settlements in Nacala. Five classes of roof material are identified: metal sheet, thatch, asbestos, concrete, and no-roof. An example of a portion of the orthophoto and roof labels is shown in Fig. C.8.

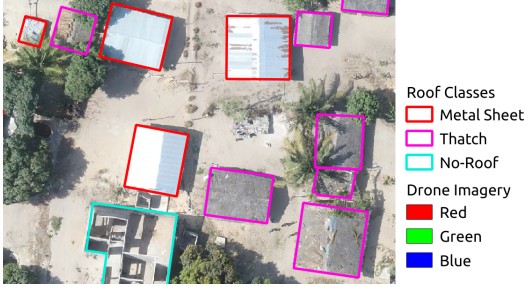

Figure C.8: Drone Imagery with RGB (Red, Green, and Blue channels) and annotations

### C.2.2 HOW MANY INSTANCES ARE THERE IN TOTAL (OF EACH TYPE, IF APPROPRIATE)?

The total number of building polygons in the data is 17954. The distribution of roof material classes is imbalanced. The number of buildings belonging to metal sheet, thatch, asbestos, concrete, and no-roof classes are 9776, 6428, 566, 174, and 1010, respectively.

### C.2.3 DOES THE DATASET CONTAIN ALL POSSIBLE INSTANCES OR IS IT A SAMPLE (NOT NECESSARILY RANDOM) OF INSTANCES FROM A LARGER SET? IF THE DATASET IS A SAMPLE, THEN WHAT IS THE LARGER SET? IS THE SAMPLE REPRESENTATIVE OF THE LARGER SET (E.G., GEOGRAPHIC COVERAGE)? IF SO, PLEASE DESCRIBE HOW THIS REPRESENTATIVENESS WAS VALIDATED/VERIFIED. IF IT IS NOT REPRESENTATIVE OF THE LARGER SET, PLEASE DESCRIBE WHY NOT (E.G., TO COVER A MORE DIVERSE RANGE OF INSTANCES, BECAUSE INSTANCES WERE WITHHELD OR UNAVAILABLE)

The dataset contains all available instances. All informal settlements in Nacala that have drone orthophotos available are prepared as a dataset. Furthermore, all buildings visible in the orthophotos are included, and the five identified building classes cover all possible roof materials in the area, and the most predominant roof materials present in the wider Nacala region.

### C.2.4 WHAT DATA DOES EACH INSTANCE CONSIST OF? "RAW" DATA (E.G., UNPROCESSED TEXT OR IMAGES)OR FEATURES? IN EITHER CASE, PLEASE PROVIDE A DESCRIPTION

The data consists of aerial images and corresponding labels. Labels are building footprints with the attribute of roof class. The raw images are GeoTiff images tagged with a spatial reference system. The raw labels are GeoJSON files with the same spatial reference system as images.

### C.2.5 IS THERE A LABEL OR TARGET ASSOCIATED WITH EACH INSTANCE? IF SO, PLEASE PROVIDE A DESCRIPTION.

The labels on the image are polygons describing the geometry of the building footprints and their associated roof material classes, as described above. In the raw data, the material class is saved under the attribute name of **mater_id** in GeoJSON files. The values of metal sheet, thatch, asbestos, concrete, and no-roof in the attribute are 1, 2, 3, 4, and 5, respectively. The same values are assigned to the patch labels.

### C.2.6 IS ANY INFORMATION MISSING FROM INDIVIDUAL INSTANCES? IF SO, PLEASE PROVIDE A DESCRIPTION, EXPLAINING WHY THIS INFORMATION IS MISSING (E.G., BECAUSE IT WAS UNAVAILABLE). THIS DOES NOT INCLUDE INTENTIONALLY REMOVED INFORMATION BUT MIGHT INCLUDE, E.G., REDACTED TEXT.

Everything is included. No data is missing.

### C.2.7 ARE RELATIONSHIPS BETWEEN INDIVIDUAL INSTANCES MADE EXPLICIT (E.G., USERS' MOVIE RATINGS, SOCIAL NETWORK LINKS)? IF SO, PLEASE DESCRIBE HOW THESE RELATIONSHIPS ARE MADE EXPLICIT.

No, the geometry and material attribute of each building footprint is independently recorded.

### C.2.8 ARE THERE RECOMMENDED DATA SPLITS (E.G., TRAINING, DEVELOPMENT/VALIDATION, TESTING)? IF SO, PLEASE PROVIDE A DESCRIPTION OF THESE SPLITS, EXPLAINING THE RATIONALE BEHIND THEM.

The roof material classes are not balanced and are not geographically distributed uniformly. The data is split into training, validation, and test sets using stratified random sampling to account for the class imbalance. We created a square grid of 225 meters and counted the roof types in these cells. Then we partitioned the cells into three sets based on the class counts to achieve a similar class distribution in each dataset, where we prioritized the distribution of minority classes (i.e., concrete and asbestos). We defined that a building only belongs to a specific grid cell if its centroid falls into the cell. These grid cells separate the images and labels into training, validation and test sets. See Fig. C.9 as an example. Initially, only two informal settlements were labelled and therefore, only these two settlements are divided into the 3 sets. The third informal settlement was labelled later and treated as a second test set.

### C.2.9 ARE THERE ANY ERRORS, SOURCES OF NOISE, OR REDUNDANCIES IN THE DATASET? IF SO, PLEASE PROVIDE A DESCRIPTION.

The images exhibit a high level of details and the building footprint geometry and material attributes are meticulous noted by experts. The dataset is free from errors, noise, or redundancies to the greatest extent possible but we acknowledge that even with expert craftsmanship, there is always a chance of human error.

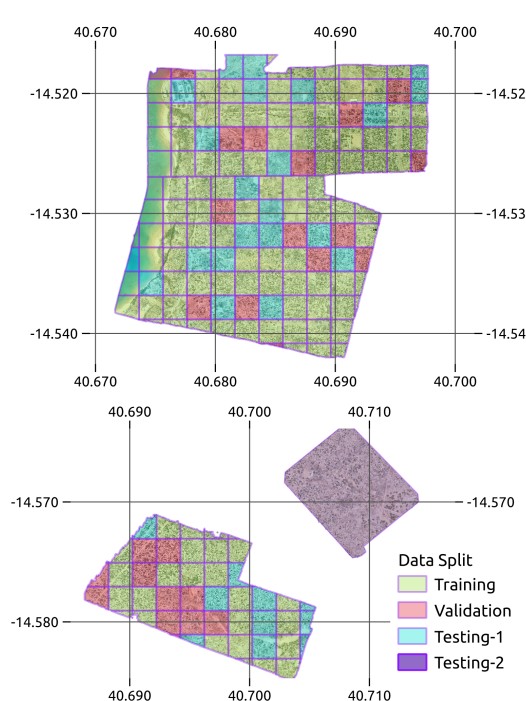

Figure C.9: Visualization of the training, validation and testing sets with reference to longitude and latitude

C.2.10 IS THE DATASET SELF-CONTAINED, OR DOES IT LINK TO OR OTHERWISE RELY ON EXTERNAL RESOURCES (E.G., WEBSITES, TWEETS, OTHER DATASETS)? IF IT LINKS TO OR RELIES ON EXTERNAL RESOURCES, A) ARE THERE GUARANTEES THAT THEY WILL EXIST, AND REMAIN CONSTANT, OVER TIME; B) ARE THERE OFFICIAL ARCHIVAL VERSIONS OF THE COMPLETE DATASET (I.E., INCLUDING THE EXTERNAL RESOURCES AS THEY EXISTED AT THE TIME THE DATASET WAS CREATED); C) ARE THERE ANY RESTRICTIONS (E.G., LICENSES, FEES) ASSOCIATED WITH ANY OF THE EXTERNAL RESOURCES THAT MIGHT APPLY TO A FUTURE USER? PLEASE PROVIDE DESCRIPTIONS OF ALL EXTERNAL RESOURCES AND ANY RESTRICTIONS ASSOCIATED WITH THEM, AS WELL AS LINKS OR OTHER ACCESS POINTS, AS APPROPRIATE.

The dataset is entirely self-contained.

C.2.11 DOES THE DATASET CONTAIN DATA THAT MIGHT BE CONSIDERED CONFIDENTIAL (E.G., DATA THAT IS PROTECTED BY LEGAL PRIVILEGE OR BY DOCTOR−PATIENT CONFIDENTIALITY, DATA THAT INCLUDES THE CONTENT OF INDIVIDUALS' NONPUBLIC COMMUNICATIONS)? IF SO, PLEASE PROVIDE A DESCRIPTION.

The dataset does not contain any confidential data.

C.2.12 DOES THE DATASET CONTAIN DATA THAT, IF VIEWED DIRECTLY, MIGHT BE OFFENSIVE, INSULTING, THREATENING, OR MIGHT OTHERWISE CAUSE ANXIETY? IF SO, PLEASE DESCRIBE WHY.

No.

C.2.13 DOES THE DATASET IDENTIFY ANY SUBPOPULATIONS (E.G., BY AGE, GENDER)? IF SO, PLEASE DESCRIBE HOW THESE SUBPOPULATIONS ARE IDENTIFIED AND PROVIDE A DESCRIPTION OF THEIR RESPECTIVE DISTRIBUTIONS WITHIN THE DATASET.

No.

### C.2.14 Is it possible to identify individuals (i.e., one or more natural persons), either directly or indirectly (i.e., in combination with other data) from the dataset? If so, please describe how.

It is not possible to identify individuals in the drone imagery. In any publicly available and geo-coded image, it is possible to identify individual houses and reverse geocode into a human-readable address.

### C.2.15 Does the dataset contain data that might be considered sensitive in any way (e.g., data that reveals race or ethnic origins, sexual orientations, religious beliefs, political opinions or union memberships, or locations; financial or health data; biometric or genetic data; forms of government identification, such as social security numbers; criminal history)? If so, please provide a description

No.

### C.2.16 Any other comments?

None.

## C.3 Collection Process

### C.3.1 How was the data associated with each instance acquired? Was the data directly observable (e.g., raw text, movie ratings), reported by subjects (e.g., survey responses), or indirectly inferred/derived from other data (e.g., part-of-speech tags, model-based guesses for age or language)? If data was reported by subjects or indirectly inferred/derived from other data, was the data validated/verified? If so, please describe how.

The data is observable as images. The ground sampling distance of pixel or spatial resolution of the imagery is $\approx 4.4$ cm/pixel. QGIS was used for the visualization of images and re-registration, cleaning, and verification of building footprints and their attributes.

### C.3.2 What mechanisms or procedures were used to collect the data (e.g., hardware apparatus or sensor, manual human curation, software program, software API)? How were these mechanisms or procedures validated?

The drone imagery was captured using a DJI Phantom 4 Pro drone and processed using AgiSoft Metashape software. All building footprints are annotated using Open Street Map (JOSM) that use OpenStreetMap in the backend. Before splitting into train, validation, and test sets, the missing labels and all geometric and attribute errors were corrected in QGIS software.

Table C.10: Drone flight information summary

| | |
|---|---|
| Number of flights: | 4 |
| Drone: | DJI Phantom 4 Pro |
| Camera Brand: | DJI |
| Camera Model: | FC6310 |
| Image Resolution: | $4864 \times 3648$ ($\sim$18MP) |
| Flight Altitude: | 120 m |
| Flight dates: | 27-30, October 2021 |
| Total flight duration: | 504 minutes |

### C.3.3 IF THE DATASET IS A SAMPLE FROM A LARGER SET, WHAT WAS THE SAMPLING STRATEGY (E.G., DETERMINISTIC, PROBABILISTIC WITH SPECIFIC SAMPLING PROBABILITIES)?

The data was prepared based on its availability. All data from the project was made available and used in this dataset.

### C.3.4 WHO WAS INVOLVED IN THE DATA COLLECTION PROCESS (E.G., STUDENTS, CROWDWORKERS, CONTRACTORS) AND HOW WERE THEY COMPENSATED (E.G., HOW MUCH WERE CROWDWORKERS PAID)?

The drone imagery was captured by an NGO. Nacala residents and local university students performed the field data collection, receiving stipends and data bundles. The polygons and attributes of building footprints were corrected by authors.

### C.3.5 OVER WHAT TIMEFRAME WAS THE DATA COLLECTED? DOES THIS TIMEFRAME MATCH THE CREATION TIMEFRAME OF THE DATA ASSOCIATED WITH THE INSTANCES (E.G., RECENT CRAWL OF OLD NEWS ARTICLES)? IF NOT, PLEASE DESCRIBE THE TIMEFRAME IN WHICH THE DATA ASSOCIATED WITH THE INSTANCES WAS CREATED.

All drone imagery was captured between October and December 2021. All labels are manually annotated on the imagery beginning January 2022 until May 2024.

### C.3.6 WERE ANY ETHICAL REVIEW PROCESSES CONDUCTED (E.G., BY AN INSTITUTIONAL REVIEW BOARD)? IF SO, PLEASE PROVIDE A DESCRIPTION OF THESE REVIEW PROCESSES, INCLUDING THE OUTCOMES, AS WELL AS A LINK OR OTHER ACCESS POINT TO ANY SUPPORTING DOCUMENTATION.

No ethical review was conducted. All data collection, including drone flights and on-site mapping, was approved and supported by the Nacala Municipal Council and facilitated on the ground by neighbourhood-level authorities.

### C.3.7 DID YOU COLLECT THE DATA FROM THE INDIVIDUALS IN QUESTION DIRECTLY, OR OBTAIN IT VIA THIRD PARTIES OR OTHER SOURCES (E.G., WEBSITES)?

The data was not collected from individuals.

### C.3.8 WERE THE INDIVIDUALS IN QUESTION NOTIFIED ABOUT THE DATA COLLECTION? IF SO, PLEASE DESCRIBE (OR SHOW WITH SCREENSHOTS OR OTHER INFORMATION) HOW NOTICE WAS PROVIDED, AND PROVIDE A LINK OR OTHER ACCESS POINT TO, OR OTHERWISE REPRODUCE, THE EXACT LANGUAGE OF THE NOTIFICATION ITSELF.

N/A.

### C.3.9 DID THE INDIVIDUALS IN QUESTION CONSENT TO THE COLLECTION AND USE OF THEIR DATA? IF SO, PLEASE DESCRIBE (OR SHOW WITH SCREENSHOTS OR OTHER INFORMATION) HOW CONSENT WAS REQUESTED AND PROVIDED, AND PROVIDE A LINK OR OTHER ACCESS POINT TO, OR OTHERWISE REPRODUCE, THE EXACT LANGUAGE TO WHICH THE INDIVIDUALS CONSENTED.

N/A.

### C.3.10 IF CONSENT WAS OBTAINED, WERE THE CONSENTING INDIVIDUALS PROVIDED WITH A MECHANISM TO REVOKE THEIR CONSENT IN THE FUTURE OR FOR CERTAIN USES? IF SO, PLEASE PROVIDE A DESCRIPTION, AS WELL AS A LINK OR OTHER ACCESS POINT TO THE MECHANISM (IF APPROPRIATE).

N/A.

### C.3.11 HAS AN ANALYSIS OF THE POTENTIAL IMPACT OF THE DATASET AND ITS USE ON DATA SUBJECTS (E.G., A DATA PROTECTION IMPACT ANALYSIS) BEEN CONDUCTED? IF SO, PLEASE PROVIDE A DESCRIPTION OF THIS ANALYSIS, INCLUDING THE OUTCOMES, AS WELL AS A LINK OR OTHER ACCESS POINT TO ANY SUPPORTING DOCUMENTATION.

N/A.

### C.3.12 ANY OTHER COMMENTS?

None.

## C.4 PREPROCESSING/CLEANING/LABELING

### C.4.1 WAS ANY PREPROCESSING/CLEANING/LABELING OF THE DATA DONE (E.G., DISCRETIZATION OR BUCKETING, TOKENIZATION, PART-OF-SPEECH TAGGING, SIFT FEATURE EXTRACTION, REMOVAL OF INSTANCES, PROCESSING OF MISSING VALUES)? IF SO, PLEASE PROVIDE A DESCRIPTION. IF NOT, YOU MAY SKIP THE REMAINDER OF THE QUESTIONS IN THIS SECTION.

Because of the large size of raw aerial imagery, the images of training and validation sets were cropped to $512 \times 512$ pixels of **patches**. The data processing optimized is usefulness in training deep learning models. The total number of patches in the training and validation sets are 8366 and 1799, respectively, after cropping them without any overlap. The test sets were provided without cropping into patches, so these images are provided in different sizes. The images in test sets are not cropped because dividing them into patches may lead to under- or over-estimation of instances and may influence counting accuracy over large areas. The test-1 set consists of 22 images and the test-2 set consists of a single image. There are 10930, 2956, 2527 and 1541 buildings in train, validation, test-1 and test-2 sets, respectively.

For the classification of buildings into different roof classes, the DINOv2 features were extracted for train, validation and test-1 sets and made available with the dataset. These features of the train, validation, and test-1 buildings are saved in train.npy, test.npy and valid.npy files, respectively. Each row in the NumPy file is a DINOv2 feature of a single building along with a label in its last column. The five roof classes are there in the data: 1-Metal Sheet, 2-Thatch, 3-Asbestos, 4-Concrete and 5-No Roof. The feature extraction is further explained in our research paper.

### C.4.2 WAS THE "RAW" DATA SAVED IN ADDITION TO THE PREPROCESSED/CLEANED/LABELED DATA (E.G., TO SUPPORT UNANTICIPATED FUTURE USES)? IF SO, PLEASE PROVIDE A LINK OR OTHER ACCESS POINT TO THE "RAW" DATA.

Yes. Along with patches and their labels, the dataset contains the raw data.

### C.4.3 IS THE SOFTWARE USED TO PREPROCESS/CLEAN/LABEL THE INSTANCES AVAILABLE? IF SO, PLEASE PROVIDE A LINK OR OTHER ACCESS POINT.

The Python script is used to prepare patches and label different deep-learning models (e.g., UNet and YOLOv8). The Python script is available in the provided zip files.

### C.4.4 ANY OTHER COMMENTS?

None.

## C.5 USES

### C.5.1 HAS THE DATASET BEEN USED FOR ANY TASKS ALREADY? IF SO, PLEASE PROVIDE A DESCRIPTION

At the time of preparing this datasheet, the dataset was only used for tasks performed in our paper.

### C.5.2 IS THERE A REPOSITORY THAT LINKS TO ANY OR ALL PAPERS OR SYSTEMS THAT USE THE DATASET? IF SO, PLEASE PROVIDE A LINK OR OTHER ACCESS POINT

No.

### C.5.3 WHAT (OTHER) TASKS COULD THE DATASET BE USED FOR?

There are other objects in the images that can also be mapped, for example, trees, roads, water bodies, etc.

### C.5.4 IS THERE ANYTHING ABOUT THE COMPOSITION OF THE DATASET OR THE WAY IT WAS COLLECTED AND PREPROCESSED/CLEANED/LABELED THAT MIGHT IMPACT FUTURE USES? FOR EXAMPLE, IS THERE ANYTHING THAT A DATASET CONSUMER MIGHT NEED TO KNOW TO AVOID USES THAT COULD RESULT IN UNFAIR TREATMENT OF INDIVIDUALS OR GROUPS (E.G., STEREOTYPING, QUALITY OF SERVICE ISSUES) OR OTHER RISKS OR HARMS (E.G., LEGAL RISKS, FINANCIAL HARMS)? IF SO, PLEASE PROVIDE A DESCRIPTION. IS THERE ANYTHING A DATASET CONSUMER COULD DO TO MITIGATE THESE RISKS OR HARMS?

There is no risk of using this dataset.

### C.5.5 ARE THERE TASKS FOR WHICH THE DATASET SHOULD NOT BE USED? IF SO, PLEASE PROVIDE A DESCRIPTION.

None.

### C.5.6 ANY OTHER COMMENTS

None.

## C.6 DISTRIBUTION

### C.6.1 WILL THE DATASET BE DISTRIBUTED TO THIRD PARTIES OUTSIDE OF THE ENTITY (E.G., COMPANY, INSTITUTION, ORGANIZATION) ON BEHALF OF WHICH THE DATASET WAS CREATED? IF SO, PLEASE PROVIDE A DESCRIPTION.

Currently, the dataset is made available through this anonymous url: `https://osf.io/us628/?view_only=3c25a48d420f4ec7a43cb76e66e92b26`. But later, we will publish through our webpage.

### C.6.2 HOW WILL THE DATASET WILL BE DISTRIBUTED (E.G., TARBALL ON WEBSITE, API, GITHUB)? DOES THE DATASET HAVE A DIGITAL OBJECT IDENTIFIER (DOI)?

Same as above answer.

### C.6.3 WHEN WILL THE DATASET BE DISTRIBUTED?

We will distribute our dataset through other sources as soon as our paper is accepted.

### C.6.4 WILL THE DATASET BE DISTRIBUTED UNDER A COPYRIGHT OR OTHER INTELLECTUAL PROPERTY (IP) LICENSE, AND/OR UNDER APPLICABLE TERMS OF USE (TOU)? IF SO, PLEASE DESCRIBE THIS LICENSE AND/OR TOU, AND PROVIDE A LINK OR OTHER ACCESS POINT TO, OR OTHERWISE REPRODUCE, ANY RELEVANT LICENSING TERMS OR TOU, AS WELL AS ANY FEES ASSOCIATED WITH THESE RESTRICTIONS

The dataset will be released under Open Data Commons Open Database License (ODbL) v1.0 licence.

C.6.5 HAVE ANY THIRD PARTIES IMPOSED IP-BASED OR OTHER RESTRICTIONS ON THE DATA ASSOCIATED WITH THE INSTANCES? IF SO, PLEASE DESCRIBE THESE RESTRICTIONS, AND PROVIDE A LINK OR OTHER ACCESS POINT TO, OR OTHERWISE REPRODUCE, ANY RELEVANT LICENSING TERMS, AS WELL AS ANY FEES ASSOCIATED WITH THESE RESTRICTIONS.

No

C.6.6 DO ANY EXPORT CONTROLS OR OTHER REGULATORY RESTRICTIONS APPLY TO THE DATASET OR TO INDIVIDUAL INSTANCES? IF SO, PLEASE DESCRIBE THESE RESTRICTIONS, AND PROVIDE A LINK OR OTHER ACCESS POINT TO, OR OTHERWISE REPRODUCE, ANY SUPPORTING DOCUMENTATION

No

C.6.7 ANY OTHER COMMENTS?

None.

## C.7 DATASET MAINTENANCE

C.7.1 WHO IS SUPPORTING/HOSTING/MAINTAINING THE DATASET?

Two of our authors will be responsible for hosting and maintaining our dataset. We will add more details here as soon as our paper accepted.

C.7.2 HOW CAN THE OWNER/CURATOR/MANAGER OF THE DATASET BE CONTACTED (E.G., EMAIL ADDRESS)?

Curators of this dataset can be communicated through our email addresses. We will provide more details as soon as our paper is accepted.

C.7.3 IS THERE AN ERRATUM? IF SO, PLEASE PROVIDE A LINK OR OTHER ACCESS POINT.

No, this is the initial release.

C.7.4 WILL THE DATASET BE UPDATED (E.G., TO CORRECT LABELING ERRORS, ADD NEW INSTANCES, DELETE INSTANCES)? IF SO, PLEASE DESCRIBE HOW OFTEN, BY WHOM, AND HOW UPDATES WILL BE COMMUNICATED TO DATASET CONSUMERS (E.G., MAILING LIST, GITHUB)?

In case of any updates, we will communicate through our webpage (we will provide the link later).

C.7.5 IF THE DATASET RELATES TO PEOPLE, ARE THERE APPLICABLE LIMITS ON THE RETENTION OF THE DATA ASSOCIATED WITH THE INSTANCES (E.G., WERE THE INDIVIDUALS IN QUESTION TOLD THAT THEIR DATA WOULD BE RETAINED FOR A FIXED PERIOD OF TIME AND THEN DELETED)? IF SO, PLEASE DESCRIBE THESE LIMITS AND EXPLAIN HOW THEY WILL BE ENFORCED.

N/A.

C.7.6 WILL OLDER VERSIONS OF THE DATASET CONTINUE TO BE SUPPORTED/HOSTED/MAINTAINED? IF SO, PLEASE DESCRIBE HOW. IF NOT, PLEASE DESCRIBE HOW ITS OBSOLESCENCE WILL BE COMMUNICATED TO DATASET CONSUMERS. THE DATASET HAS ALREADY BEEN UPDATED; OLDER VERSIONS ARE KEPT AROUND FOR CONSISTENCY

N/A.

### C.7.7 IF OTHERS WANT TO EXTEND/AUGMENT/BUILD ON THIS DATASET, IS THERE A MECHANISM FOR THEM TO DO SO? IF SO, IS THERE A PROCESS FOR TRACKING/ASSESSING THE QUALITY OF THOSE CONTRIBUTIONS. WHAT IS THE PROCESS FOR COMMUNICATING/DISTRIBUTING THESE CONTRIBUTIONS TO USERS?

N/A

### C.7.8 ANY OTHER COMMENTS?

None.

