# OpenReview forum: "Nacala-Roof-Material: Drone Imagery for Roof Detection, Classification, and Segmentation to Support Mosquito-borne Disease Risk Assessment"
_ICLR.cc/2025/Conference — ICLR 2025 Conference Withdrawn Submission_

### Official Review · Reviewer_E5bP · 2024-11-02

**Soundness:** 3
**Presentation:** 3
**Contribution:** 2
**Rating:** 5
**Confidence:** 3

**Summary:**

This paper presented a root material segmentation and detection dataset based on drone images. Authors also benchmarked YOLOv8, Unet, DINOv2 models on their dataset at both pixel-level and object-level.

**Strengths:**

1. The roof material segmentation and detection dataset is valuable for autormatic house quality estimation.
2. The paper is writtn well and easy to follow.
3. The visual foundation model, e.g., DINOv2, is considered into their study.

**Weaknesses:**

1. The benchmark is not comprehensive. There are many good object detection, semantic segmentation, and instance segmentation models can be evaluated on this new dataset for more insights.
2. The analysis is insufficient, e.g., what are the main error sources in this task? how well does the larger vision foundation models (e.g., SAM) do? The authors should give more insights to guide future readers in exploring roof material classification problems.
3. A comparison between conventional datasets and your dataset should be included.

**Questions:**

see Weaknesses

---

### Official Review · Reviewer_EgFJ · 2024-11-03

**Soundness:** 3
**Presentation:** 3
**Contribution:** 1
**Rating:** 1
**Confidence:** 4

**Summary:**

The paper proposes a dataset to classify and segment roof materials motivated by a correlation of certain roof materials with mosquito risks. It is certainly an interesting dataset and problem. The dataset is certainly impactful, as it has been created with substantial manual effort by local university students and NGO members from Nacala. Also, the distinction between a spatial in-distribution D_text and spatially distinct D_ext, some classes missing in the training sets, is certainly a challenging setup.
However, I evaluate the impact of this paper for a machine learning conference is limited. In terms of methodology, the paper uses existing (pre-trained) vision models and proposes a watershed postprocessing step, "DOW," inspired by  Cheng et al. (2024). In my opinion, it would fit better to an ICLR workshop (e.g., Machine Learning for Remote Sensing) or in a Remote Sensing Journal/Conference where the impact of the paper could be better evaluated and where it also would have more impact overall.

**Strengths:**

It is an interesting problem and a well-curated dataset with several existing baseline models

**Weaknesses:**

* My central concern is the limited relevance of this dataset to a machine learning audience, as it is mainly relevant to remote sensing researchers. I did not see any aspects of this dataset that would open a particular targeted challenge/problem to develop new machine learning models on. It's contribution is rather in the environmental/social impact than building the basis for new model developments. Hence, it would fit a dedicated workshop or a remote sensing journal better where it can be evaluated properly.

**Questions:**

The concrete link between mosquitos and roof materials did not become clear to me: Are there some studies relating exactly these two data sources, or is roof material mainly a proxy for wealth and slum mapping?
Concretely, is the connections with mosquitos merely motivational, or do you have data collected (e.g., through surveys or disease occurrence rates) that supports this connection?

---

### Official Review · Reviewer_rf69 · 2024-11-03

**Soundness:** 2
**Presentation:** 3
**Contribution:** 2
**Rating:** 3
**Confidence:** 5

**Summary:**

This paper presents the Nacala-Roof-Material dataset, which includes drone imagery from a sub-Saharan African region, designed for building segmentation and roof-type classification. The authors apply a deep ordinal watershed model to accomplish this task, with potential applications in risk assessments for vector-borne diseases.

**Strengths:**

1.This paper is written in a clear and concise way, making it enjoyable to read and easy to digest.
2. The topic is interesting and valuable.

**Weaknesses:**

1. Dataset. The design of the extend test dataset is inadequate, which does not contain any example of two roof types.
2. Related works. Missing discussion on relevant methods, such as multi-task/multi-resolution building extraction techniques.
3. Technical novelty. The approach relies heavily on established methods.

**Questions:**

I appreciate the writing style. However, due to the lack of insights and innovation in the method, submission to a remote sensing journal is suggested.

---

### Official Review · Reviewer_XaQQ · 2024-11-08

**Soundness:** 4
**Presentation:** 3
**Contribution:** 2
**Rating:** 5
**Confidence:** 4

**Summary:**

This paper introduces the Nacala-Roof-Material dataset, a high-resolution drone imagery dataset collected from informal settlements in Nacala, Mozambique. The dataset is designed to assist in research on roof detection, classification, and segmentation, with a specific focus on mitigating mosquito-borne disease risks, such as malaria, which are influenced by housing quality and roof materials. The authors define three primary tasks using this dataset: building detection, multi-class roof type classification, and pixel-level building segmentation. Several state-of-the-art models, including U-Net, YOLOv8, and DINOv2, are benchmarked to evaluate performance on these tasks. Additionally, the paper proposes the Deep Ordinal Watershed (DOW) method, which improves segmentation accuracy by separately delineating building exteriors and interiors. By releasing this dataset and associated benchmarks, the authors aim to advance research on using remote sensing for public health applications.

**Strengths:**

__Significance and Novelty:__

The paper addresses a crucial public health challenge by linking housing conditions with mosquito-borne disease risk assessment. This intersection of computer vision and epidemiology is novel and holds significant societal impact, especially in resource-limited regions. By providing a high-resolution drone imagery dataset focused on roof material classification, the research fills a gap left by existing lower-resolution datasets that do not distinguish between specific roof types.

__Soundness of the Claims:__

Theoretical grounding is evident as the authors effectively combine concepts from computer vision with epidemiological insights. The benchmarked models are carefully selected, and the inclusion of the DOW method demonstrates a thoughtful approach to improving segmentation outcomes. The experimental setup is robust, with a detailed analysis of various metrics, including intersection over union (IoU) and AP scores, ensuring empirical validity.

__Empirical Evaluation:__

The extensive empirical evaluation demonstrates that no single model is superior across all tasks, providing valuable insights for future researchers. By comparing models on multiple segmentation and classification metrics, the study offers a nuanced understanding of model performance, emphasizing practical limitations and advantages of each approach. The statistical analysis adds credibility, and the publicly available dataset promotes reproducibility.

**Weaknesses:**

__Generalization Concerns:__

Although the dataset provides a meaningful resource, it is limited to a single geographic region, which may affect the generalizability of the findings. The paper does not explore how well models trained on this dataset might perform in different regions with varying architectural styles or environmental conditions, which is a crucial aspect for real-world applications.

__Limited Baseline Diversity:__

While the models benchmarked are popular, the study could benefit from including more diverse architectures or emerging models from the latest research to better assess the dataset’s full potential. Additionally, the DOW method is compared mainly against traditional segmentation methods without an extensive exploration of other advanced post-processing techniques.

__Computational Resource Requirements:__

The paper mentions that training the models requires significant computational power, which could be a barrier for adoption, particularly in the regions where this research is most needed. This limitation reduces the accessibility of the proposed methods, especially for small organizations or researchers with limited resources.

**Questions:**

1. How well would the models generalize to datasets from different geographical areas with varying building styles and materials?

2. Are there plans to extend the dataset to include more regions to enhance generalization and applicability across sub-Saharan Africa or similar environments?

3. What measures were taken to ensure that the manual annotations are free from bias, and how might annotation inconsistencies affect the benchmark results?

4. How does the DOW method compare against other state-of-the-art post-processing techniques for object delineation beyond watershed-based methods?

5. Are there specific public health interventions or organizations already interested in or collaborating on using these findings?

**Details Of Ethics Concerns:**

**A. Major Issues**

1. **Geographic Limitation:**
   The dataset is collected from a single region in Nacala, Mozambique, which might not represent roof materials or building styles across other malaria-endemic areas. This could limit the generalization and scalability of the models to diverse settings.

2. **Computational Resources:**
   The training and testing of models require considerable computational power, which might not be feasible for deployment in resource-limited settings where the research is most needed. This restricts the practical use of these models and their accessibility.

3. **Model Performance Variability:**
   The results indicate that no model performs uniformly well across all tasks, suggesting limitations in current architectures for this multi-task problem. The inconsistent performance, particularly on minority roof types, could reduce the practical utility of the models.

4. **Potential for Annotation Errors:**
   Although the annotation process was rigorous, the reliance on manual annotations introduces the risk of bias or error, especially when distinguishing visually similar roof materials like concrete and asbestos.

5. **Environmental Factors:**
   The study does not account for environmental factors such as weather, lighting variations, or seasonal changes that might influence drone imagery quality, which could impact model robustness in real-world scenarios.

**B. Minor Issues**

1. **Dataset Size and Balance:**
   The dataset is relatively small, especially for certain roof types, which could introduce class imbalance issues. Expanding the dataset or using data augmentation could help alleviate this.

2. **Documentation and Usability:**
   While the data and code are made publicly available, clearer documentation on how to use the dataset and reproduce the results would improve accessibility for a broader audience.

3. **Limited Comparisons:**
   The study could benefit from comparing its methods to additional segmentation algorithms or using ensemble techniques for a more comprehensive evaluation.

---

**C. Recommendations**

1. Consider expanding the dataset to include diverse geographic regions and building styles to enhance model generalization and applicability.
2. Improve documentation and provide detailed guidelines for researchers with limited computational resources to facilitate the use of the dataset and models.
3. Explore the integration of data augmentation techniques to address class imbalance and improve model robustness.

---

### Note · Authors · 2024-11-20

**Comment:**

Thank you for your feedback. We withdraw the manuscirpt and will submit it to a remote sensing journal as suggested by some of the reviewers.

**Withdrawal Confirmation:**

I have read and agree with the venue's withdrawal policy on behalf of myself and my co-authors.